# Hierarchical Vector Quantized Transformer for Multi-class Unsupervised Anomaly Detection

**Ruiying Lu[1][†], YuJie Wu[2][†] , Long Tian[2][*] , Dongsheng Wang[3]**
**Bo Chen[3], Xiyang Liu[2], Ruimin Hu[1]**
School of Cyber Engineering[1], Software Engineering Institute[2]
National Key Laboratory of Radar Signal Processing[3]
Xidian University
`{luruiying,tianlong}@xidian.edu.cn`

## Abstract

Unsupervised image Anomaly Detection (UAD) aims to learn robust and discriminative representations of normal samples. While separate solutions per class endow expensive computation and limited generalizability, this paper focuses on building a unified framework for multiple classes. Under such a challenging setting, popular reconstruction-based networks with continuous latent representation assumption always suffer from the "identical shortcut" issue, where both normal and abnormal samples can be well recovered and difficult to distinguish. To address this pivotal issue, we propose a hierarchical vector quantized prototype-oriented Transformer under a probabilistic framework. First, instead of learning the continuous representations, we preserve the typical normal patterns as discrete iconic prototypes, and confirm the importance of Vector Quantization in preventing the model from falling into the shortcut. The vector quantized iconic prototypes are integrated into the Transformer for reconstruction, such that the abnormal data point is flipped to a normal data point. Second, we investigate an exquisite hierarchical framework to relieve the codebook collapse issue and replenish frail normal patterns. Third, a prototype-oriented optimal transport method is proposed to better regulate the prototypes and hierarchically evaluate the abnormal score. By evaluating on MVTec-AD and VisA datasets, our model surpasses the state-of-the-art alternatives and possesses good interpretability. The code is available at `https://github.com/RuiyingLu/HVQ-Trans`.

## 1 Introduction

Anomaly detection is an essential task with increasingly wide applications in various areas, such as video surveillance [1], industrial inspection [2], and medical image analysis [3]. Due to the scarcity of anomalous samples, the unsupervised anomaly detection [4–7] methods gain wide attention by modeling the distribution of normal data only, and then identify the samples deviates from the normal profile as anomalies. Common approaches follow the one-for-one scheme [8, 9] by training separate models for different classes of objects, which is time-memory-consuming for real application and uncongenial to the object class with large intra-class diversity. Recently, a newly emerging one-for-all scheme [10] tries to use a unified model to detect anomalies from all the different object classes without any fine-tuning. Modeling high-dimensional data is notoriously challenging, and the problem becomes even more difficult to capture the multi-class distribution in a unified model precisely.

---

†Equal Contribution
* Corresponding Author

37th Conference on Neural Information Processing Systems (NeurIPS 2023).

Under the unsupervised setting, a powerful approach to modeling data distribution follows the deep autoencoding frameworks, reckoning that a well-trained model with normal data will always reconstruct normal patterns regardless of the defects present in the input data. Thus, it is generally assumed that the reconstruction error will be larger for the anomalous input, making them distinguishable from the normal samples. However, this assumption may not always hold that sometimes the abnormal inputs can also be well reconstructed, which is named as "identical shortcut" issue [9–11]. Intuitively, compared to working extremely hard to learn the joint distribution, returning a direct copy of the input disregarding its content appears as a far easier solution. This phenomenon has been observed in existing researches [4, 9]. Furthermore, under the unified case, the "identical shortcut" issue becomes even more severe as the distribution of multi-class data is more complex [10]. This motivates us to enhance the discriminability of model encountering normal and anomalous samples.

Learning representations with continuous features have been the focus of many previous works [12–14]. However, these methods lack a reliable mechanism to encourage the model to induce large reconstruction error on the anomalies, restricting the performances by the under-designed representation of the latent space. In recent researches, a branch of approaches [8, 9, 15] investigate the memory-augmented networks for mitigating the "identical shortcut" issue of AEs. Those approaches augment the deep autoencoder with a memory module to record the normal patterns in the normal training data, manifesting in different forms such as the memory set in the latent space [9], the fixed Transformer value matrix in the attention layer [10], or neighborhood-aware patch-level memory bank [8]. This kind of method hopes to obtain low reconstruction error for normal samples and highlight the reconstruction error if the input is not similar to normal data, that is, an anomaly. The most relevant items in the memory are retrieved and weighted averaging all the related memory content are aggregated into the decoder for reconstruction. However, the discrete memory items are recombined and weighted averaged, falling into an unknown continuous latent space which might be distorted. Intuitively, some anomalous regions can not be reconstructed by the discrete latent memory but could be decoded from the unknown latent space. To intrinsically mitigate the problems, we aim to learn a representative and discriminative discrete latent space for anomaly detection.

To preserve the typical normal patterns in the discrete latent space, we hope to successfully model critical features that usually span many dimensions in the normal data space, as opposed to focusing or spending capacity on noise and imperceptible details. Incorporating ideas from vector quantization (VQ), we model the discrete latent space as codebooks for each category, consisting of iconic prototypes learned from normal training data. During reconstruction, we replace the original encoding features with the nearest iconic prototypes, and then decoded with a VQ-based transformer decoder to intensify the use of iconic prototypes. As a result, the abnormal data point is flipped to a normal data point, highlighted by large reconstruction errors, as shown in Fig. 1. However, the model may suffer from the codebook collapse issue [16, 17]: At some point during training, a part of latent codes in the codebook may no longer work and the modeling capacity is limited by the discrete representations, resulting in collapsed reconstruction [18].

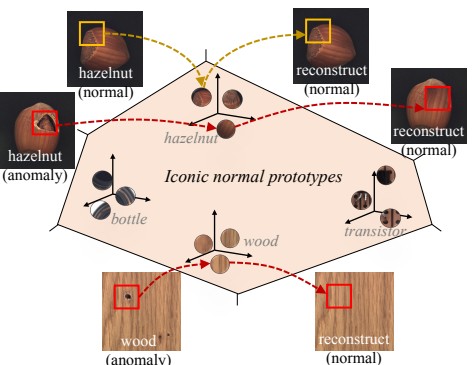

Figure 1: By replacing the continuous latent features with the normal iconic prototypes of corresponding category, the normal regions are reconstructed as normal patterns (shown in yellow boxes), while the anomalies are also reconstructed as normal (shown in red boxes).

Thus, we further investigate the hierarchical nature of images and propose a hierarchical VQ framework by merging fine-grained and abstract features to prevent codebook collapse, which could also reduce the decoding search time and retain high inference speeds. In addition, most abnormal scoring methods are constrained to the observation space and can be fallible to complex data distributions. Therefore, we have introduced a hierarchical prototype-oriented optimal transport (OT) based optimization and anomaly detection scoring method to enhance the robustness and discriminability of our model for normal and anomaly samples.

In conclusion, we carefully tailor a variational autoencoding framework for unsupervised anomaly detection, called hierarchical vector quantized Transformer (HVQ-Trans). Our work contributes in the following ways: (1) We realize the learning of discrete normal representations by extracting

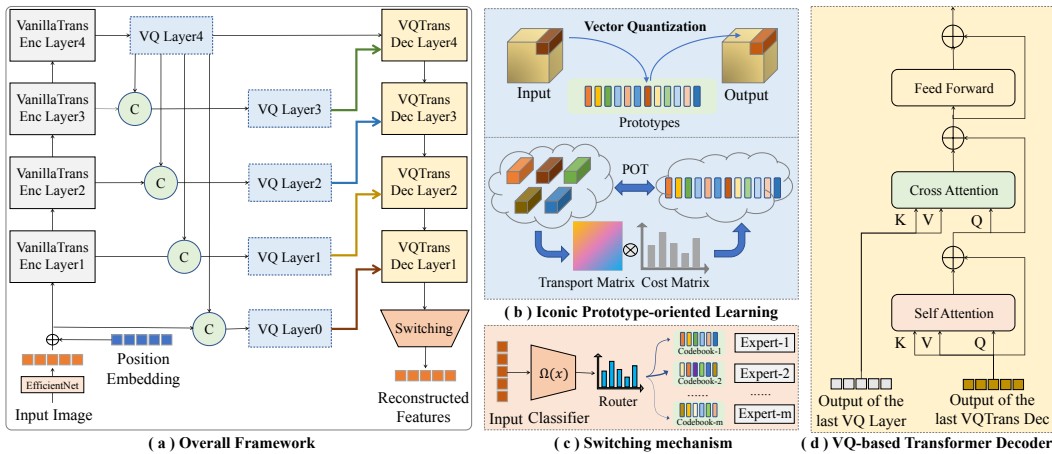

Figure 2: (a) The overall framework of our HVQ-Trans. (b) Each VQ-based Layer replaces continuous features with iconic prototypes, equipped with the POT module to promote better learning and scoring. (c) The codebook and expert network are switched for individual image. (d) The detailed structure of each VQ-based Transformer decoder, where the prototypes are integrated via cross-attention.

prototypes and propose a VQ-based transformer to address the "identical shortcut" issue by inducing large feature discrepancy for anomalies. (2) We develop a hierarchical VQ-based approach with switching mechanism to overcome the "prototype collapse" problem and effectively use multi-level feature representations to maximize the nominal information available. (3) A hierarchical prototype-oriented learning and anomaly scoring method is developed to guide prototype learning and dexterously measure the feature level anomaly score to robustly and accurately identify anomalies. (4) Extensive experiments demonstrate our method achieves state-of-the-art performances for anomaly detection and localization, and possesses enhanced interpretability through prototype visualization.

## 2 Methodology

### 2.1 Overview

Our proposed model is a Transformer-based reconstruction method that assumes normal and anomaly cannot be reconstructed with comparable performances. In contrast to other Transformer-based autoencoding [19] with continuous embeddings, we focus on compressing input images into discrete representations and achieve discriminative reconstruction for anomaly detection and localization. We denote the set of normal images available at training time as $\mathcal{X}_N$ ($\forall \boldsymbol{x} \in \mathcal{X}_N : \boldsymbol{y}_x = 0$), with $\boldsymbol{y}_x \in \{0, 1\}$ denoting if an image $\boldsymbol{x}$ is normal (0) or anomalous (1). Accordingly, we define the test sample as $\forall \boldsymbol{x} \in \mathcal{X}_T : \boldsymbol{y}_x = \{0, 1\}$, including both the normal test images and abnormal test images. The model pipeline, shown in Figure 2, can be summarized as follows: i) The input image is fed into the pre-trained EfficientNet [20] to extract visual tokens by splitting 3-D feature maps; ii) The extracted tokens are passed through the cascaded *vanilla transformer encoder* for non-local multi-level feature aggregation; iii) Aggregated features at certain layer are hierarchically fed into the corresponding *VQ-based layer* to select the most relevant iconic prototypes; iv) The visual tokens are then successively fused with vector quantized embeddings via cascaded *VQ-based transformer decoder* for reconstruction; v) Decoded tokens are passed through the *switching experts* to reconstruct features, which possess flexibility in high diversity multi-class image scenarios; vi) Finally, anomaly detection and localization are achieved through a calibrated anomaly score map refined via prototype-oriented module by measuring the OT-based hierarchical feature discrepancy.

### 2.2 Improving Feature Reconstruction with Hierarchical Vector Quantization

**Motivation:** Normal memory augmentation was initially introduced by Gong et al. [9] and has obtained wide interests in unsupervised anomaly localization and detection. To record the "normal" appearance, image features are augmented by weighted averaging the similar patterns in the memory matrix. This augmentation is, however, rebuilding a continuous latent space again which might be distorted to contain abnormal patterns. Relied on the vector quantization, discrete variational autoencoder[21] learns the discrete latent space, but suffers from the issue of codebook collapse.

Therefore, to simultaneously learn the discrete representation and avoid codebook collapse, we proposed the hierarchical VQ-based framework.

**Hierarchical Vector Quantized (HVQ) Transformer:** Our proposed HVQ-Trans can be viewed as a communication system serving as an information bottleneck to better capture the normal patterns during training, which could be further generalized to test unknown images with arbitrary anomalies. We denote the input image as $\boldsymbol{x} \in \mathbb{R}^{H \times W \times 3}$, then the $N$ visual tokens $\boldsymbol{h}^0 = f_{\boldsymbol{\phi}^0}(\boldsymbol{x}) \in \mathbb{R}^{N \times C}$ are extracted by the pre-trained EfficientNet $\phi^0$ [20] to be fed into the Transformer encoder. As shown in the graphical model of Fig. 3, HVQ-Trans comprises a cascaded vanilla Transformer encoder (*vanTrans-enc*) parameterized by $\phi^l$ that encodes multi-layer patch embeddings as $\boldsymbol{h}^l = f_{\boldsymbol{\phi}^l}(\boldsymbol{h}^{l-1}) \in \mathbb{R}^{N \times C}$. To further enlarge the normality and suppress the anomaly, we subsequently develop hierarchical VQ-based layers (*VQLayer*) to layer-wisely quantize the refined visual tokens $\{\boldsymbol{h}^l\}_{l=1}^L$ to the prototypes $\boldsymbol{e}_k^l$ in the learnable codebooks $\boldsymbol{E}^l \in \mathbb{R}^{K \times C}$, as:

$$\boldsymbol{\theta} = Quantize\left(\Upsilon^L(\boldsymbol{h}^L)\right) = \boldsymbol{e}_i^L, \quad i = \min_j \|\Upsilon^L(\boldsymbol{h}^L) - \boldsymbol{e}_j^L\|_2^2,$$

$$\boldsymbol{z}^l = Quantize\left(\Upsilon^l\left(\left[\boldsymbol{h}^{l-1}, \boldsymbol{\theta}\right]\right)\right) = \boldsymbol{e}_k^l, \quad k = \min_r \|\Upsilon^l\left(\left[\boldsymbol{h}^{l-1}, \boldsymbol{z}^L\right]\right) - \boldsymbol{e}_r^l\|_2^2, \quad l \in [1, ..., L],$$

$$(1)$$

where $\Upsilon^l(\cdot)$ refers to the layer-wise embedding function, and $[\cdot]$ denotes the concatenation operation. Intuitively, we hierarchically replace visual tokens $\boldsymbol{h}^l$ with their most similar prototypes in the codebook $\boldsymbol{E}^l$ as quantized vector $\boldsymbol{z}^l$ (note that this process can be lossy). Moreover, we find that merging fine-grained concrete information with abstraction-level semantics is critical for robust anomaly detection. Hence, we fuse the multi-level visual tokens with the global quantized vector $\boldsymbol{\theta}$ to learn hierarchical prototypes, maximizing the preserved nominal information.

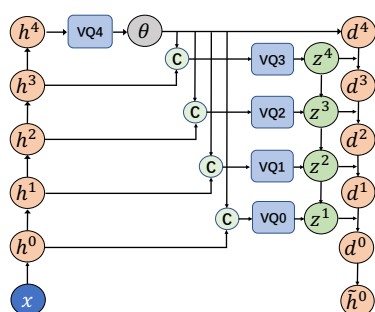

Figure 3: Graphical model of our HVQ-Trans.

To make full use of the quantized multi-level visual tokens $\boldsymbol{z}^l$ derived from by Eq. 1, we further developed a cascaded VQ-based Transformer decoder (*VQTrans-dec*). As shown in Fig. 2 (d), given the quantized visual tokens $\boldsymbol{z}^l$ of the $l^{th}$ layer and its corresponding output $\boldsymbol{d}^{l+1}$ from the last *VQTrans-dec* layer, the operation in each *VQTrans-dec* layer can be expressed as:

$$\boldsymbol{q}^l = \text{MSA}(query = \mathbf{W}_q \boldsymbol{d}^{l+1}, key = \mathbf{W}_k \boldsymbol{d}^{l+1}, value = \mathbf{W}_v \boldsymbol{d}^{l+1}) + \boldsymbol{d}^{l+1},$$

$$\tilde{\boldsymbol{d}}^l = \text{MCA}(query = \mathbf{W}_q' \boldsymbol{q}^l, key = \mathbf{W}_k' \boldsymbol{z}^l, value = \mathbf{W}_v' \boldsymbol{z}^l) + \boldsymbol{q}^l, \quad \boldsymbol{d}^l = \text{FFN}(\tilde{\boldsymbol{d}}^l) + \tilde{\boldsymbol{d}}^l, \quad (2)$$

where $\text{MSA}(\cdot)$ and $\text{MCA}(\cdot)$ share the same architecture as the standard multi-head self attention and multi-head cross attention in vanilla Transformer [22]. An essential aspect of our method is that, in the $\text{MCA}(\cdot)$ operation, the refined visual tokens from the previous layer $\boldsymbol{q}^l$ crossly attend to the prototypes of normal images $\boldsymbol{z}^l$. Hence, the values at abnormal regions of $\boldsymbol{q}^l$ will be suppressed, and the abnormal signals could be rarely transmitted to the output terminal for reconstruction. Namely, the fewer reconstructed anomalies there are, the larger the reconstruction difference will be, which in turn leads to better performance in localizing and detecting anomalies.

During training, the typical normal patterns are recorded in the discrete variables, *i.e.*, iconic prototypes. When encountering the anomalous during testing, the abnormal patterns will also be quantized as the normal prototypes, leading to larger feature migration and information loss, highlighted by higher reconstruction error. It is worth noting that while information loss triggered by VQ is exist for normal images, it is significantly more pronounced for anomaly images. This discrepancy in information loss serves as a key factor in effective anomaly detection. By investigating this difference, we can enhance the accuracy of our model in distinguishing abnormal regions.

**Switching Mechanism:** We adopt the switching mechanism to make the proposed model more suitable for multi-class anomaly detection. On the one hand, we develop the *switching codebooks* by assembling the independent iconic prototypes for each class. On the other hand, we develop the *switching experts* for flexible reconstruction of multi-classes, inspired by the Mixture of Experts (MoE) models [23, 24] and its sparsely-activated version [25]. Here, we choose to reconstruct at the

feature level rather than the pixel level, due to the invariance to subtle noise, rotation, and translation at the pixel level.

Specifically, the switching mechanism contains a multi-category classifier, $M$ codebooks, and $M$ reconstruction experts. The multi-category classifier takes the image feature as input and outputs the classification probability over the $M$ category. In order to fit the data diversity property in the one-for-all setting, we switch the specific codebook (including a group of prototypes) from $M$ codebooks according to the classification probability. Furthermore, we switch the individual reconstruction network (dubbed as expert) for feature reconstruction. The visual tokens from the last *VQTrans-dec* layer are depicted as $d^0 \in \mathbb{R}^{N \times C}$, which is expected to reconstruct the input patch features $h^0 \in \mathbb{R}^{N \times C}$ as:

$$\tilde{h}^0 = \Psi_m(d^0), \quad m = \underset{j}{\arg\max}\, p_j(x), \quad p_j(x) = \frac{exp(\Omega(x)_j)}{\sum_{j=1}^{M} exp(\Omega(x)_j)}, \tag{3}$$

where $\Omega(\cdot)$ is a classifier for producing logits, which are then normalized via a Softmax function over the total $M$ experts. $p_j(\cdot)$ is the probability of selecting the codebook and reconstruction expert, as shown in Fig. 2 (c). The codebook and expert $\Psi_m$ with the highest probability are employed for reconstruction. The switching mechanism under the one-for-all setting will classify each input image into a single category and choose the corresponding codebook and expert for reconstruction. For the normal images, it is highly likely to be classified into the correct category and thus switch the proper reconstruction expert and codebook. For the abnormal images, there remains big uncertainty that which reconstruction expert and codebook will be switched, because the anomalies are unseen during training. Thus, the reconstruction uncertainty of the abnormal image is increased. Noting that the difference between normal and abnormal is the key factor deciding the anomaly detection performance. To this end, the uncertainty during anomalous sample switching could facilitate multi-class anomaly detection.

## 2.3 Prototype-oriented Learning and Scoring for Anomaly Detection

**Motivation:** During training with normal data, the HVQ-Trans enhances the point-wise correlations of the selected iconic prototypes from codebooks and the continuous visual features of normal images. However, it may despise the global relations between the above two sets. Motivated by previous efforts on OT theory and applications [26–29], we propose a hierarchical prototype-oriented optimal transport (POT) for anomaly detection, which is a transport solver defined within the scope of the basic distance between two unknown sampling sets to improve the tightness between the codebooks and the normal features. Meanwhile, at the testing stage, it is worth noting that the anomaly score adopted in the existing methods [10, 17] mainly concern the most significant difference of the score map between the input features and the reconstructed ones, as measured by the Euclidean distance. However, the importance of hierarchical differences at multiple feature levels is neglected. To this end, we also proposed a hierarchical POT-based anomaly scoring method to reinforce the identification of the score map and further boost the anomaly detection performance.

**Learning Iconic Prototypes with POT:** Each POT module, included within each *VQLayer* as shown in Fig. 2 (b), is responsible for enhancing consistency between codebooks and normal image features per layer. This enables the prototypes in codebooks to be more representative of normal patterns and less so of anomalous patterns. At the $l$-th layer, the codebook of each category contains a group of prototypes $e^l = [e_1^l, ..., e_K^l] \in \mathbb{R}^{K \times C}$. We omit the index $l$ in the following for simplicity without causing ambiguity. To assemble the normal patterns conveyed by images, we represent $N$ patches per image as an empirical distribution $\mathbb{P}_h = \sum_{i=1}^{N} \frac{1}{N} \delta_{h_i}$, where $h \in \mathbb{R}^{N \times C}$ is the features sampled from the latent variables of the HVQ-Trans. The prototypes serve to represent normal patterns across different classes. As a result, when attempting to identify suitable prototypes to reconstruct a specific normal image, each prototype is given equal importance. Thus, the distribution over normal prototypes can also be expressed as an empirical distribution $\mathbb{P}_e = \sum_{j=1}^{K} \frac{1}{K} \delta_{e_j}$. In this way, the transport matrix $M^* \in \mathbb{R}^{N \times K}$ from $\mathbb{P}_h$ to $\mathbb{P}_e$ can be estimated by $M^* = \min_{M} \sum_{i=1}^{N} \sum_{j=1}^{K} M_{i,j} C_{i,j}$, where the transport matrix $M$ should satisfy $\Pi([\frac{1}{K}], [\frac{1}{N}]) = \{M | M\mathbf{1}_K = [\frac{1}{K}], M^T\mathbf{1}_N = [\frac{1}{N}]\}$. $[\frac{1}{K}]$ and $[\frac{1}{N}]$ are two uniform distributed prior defined in $\mathbb{P}_h$ and $\mathbb{P}_e$, respectively. The cost matrix $C \in \mathbb{R}^{N \times K}$ is defined as $C_{i,j} = \sqrt{(h_i - e_j)^2}$. In order to learn the prototypes of normal codebook at certain layer,

we define the average POT loss inspired by Sinkhorn algorithm [30] as:

$$\mathbb{L}_{POT} = \min_{\boldsymbol{E}} \mathbb{E}_{\boldsymbol{h} \sim F_{\phi}(\boldsymbol{x})} \sum_{i=1}^{N} \sum_{j=1}^{K} \boldsymbol{M}_{i,j}^{*} \boldsymbol{C}_{i,j} + \sum_{i=1}^{N} \sum_{j=1}^{K} \boldsymbol{M}_{i,j}^{*} ln \boldsymbol{M}_{i,j}^{*}. \tag{4}$$

**Calibrating Anomaly Score with POT:** The anomaly score computed via Transformer-based methods always suffers from the sub-optimal distance measurement, which is usually calculated as the point-wise L2 norm of the reconstruction differences as $\boldsymbol{s}_{org} = \|\boldsymbol{f}_{org} - \boldsymbol{f}_{rec}\|_2^2$. In this paper, on the one hand, we alleviate the mismatch by restricting the distance of prototypes and visual features during training. On the other hand, we further calibrate the anomaly score with multi-level POT at test time. Accordingly, in our proposed method, we note that the anomaly degree could also be reflected by the dissimilarity between visual features and normal iconic prototypes in the codebooks. As for the $l$-th layer, the dissimilarity is evaluated by $\boldsymbol{s}_{POT}^{l} = \boldsymbol{M}^{*} \boldsymbol{C}$. The transport matrix $\boldsymbol{M}^{*}$ acts as a probability to re-weight the cost with different prototypes $\boldsymbol{C}$, which measures the importance of different distances between image features and the normal prototypes. Therefore, we calibrate the multi-level anomaly score as $\boldsymbol{s}_{cab} = \boldsymbol{s}_{org} + \lambda \sum_{l=1}^{L} \boldsymbol{s}_{POT}^{l}$ for better anomaly detection.

## 2.4 Overall Optimization

Noting that there is no real gradient defined for equation 1, following [17, 21, 31, 32], we approximate the gradient by copying gradients from the refined visual tokens $z^l$ to the visual tokens $h^l$ for $l = 0, ..., L$. Thus, our proposed HVQ-Trans incorporates five terms into its objective, specified as:

$$\mathbb{L}_{HVQ-Trans} = ||\boldsymbol{h}^0 - \tilde{\boldsymbol{h}}^0||_2^2 + \sum_{l=1}^{L} \left[ ||\text{sg}(\boldsymbol{h}^l) - \boldsymbol{e}^l||_2^2 + \beta^l ||\boldsymbol{h}^l - \text{sg}(\boldsymbol{e}^l)||_2^2 + \alpha^l \mathbb{L}_{POT}^l \right] - \sum_{j=1}^{M} p_j(\boldsymbol{x}) \log \mathcal{P}_x, \tag{5}$$

where the $\text{sg}(\cdot)$ refers to the stop-gradient operation and $\mathcal{P}_x$ the category label. $\beta^l$ and $\alpha^l$ are hyperparameters. The first term refers to the reconstruction loss. The second one in the scope of summation along $L$ layers is the hierarchical prototypical loss, pushing the selected prototype $\boldsymbol{e}^l$ closer to the visual token $\boldsymbol{h}^l$. The third term denotes the hierarchical commitment loss, optimizing the encoder by encouraging the output of the encoder $\boldsymbol{h}^l$ to stay close to the chosen prototype and prevent it from fluctuating too frequently from one prototype to another. The fourth term is the POT loss defined in Eq. 4. The last term is the cross entropy loss for training the classifier to adaptively switch the proper reconstruction expert and codebook. Following [21], we use the exponential moving average updates for codebooks. More details can be found in Appendix.

## 3 Connection with previous works

In unsupervised anomaly detection, only normal samples are available at the training stage. Unsupervised anomaly detection methods can be roughly categorized into density-based and reconstruction-based methods. Density-based methods estimate the distribution of normal data points to identify anomalous data points [4–6]. Our HVQ-Trans method adheres to the probabilistic variation framework but refrains from assuming a specific distribution of normal data. Therefore, the image prior is learned dynamically rather than relying on a static distribution. On the other hand, reconstruction-based methods assume that the model trained on normal data only can well reconstruct normal regions, but fail in anomalous regions [33–35]. Typical approaches include Auto-Encoder (AE) [36–38], Variational Auto-Encoder (VAE) [39, 40], and Generative Adversarial Net (GAN) [41–43]. However, most of these methods do not incorporate a reliable mechanism for encouraging the model to induce large reconstruction error on the anomalous region.

Adopting a memory matrix for unsupervised anomaly detection has proven to be an effective solution. The idea was first proposed in MemAE [9] by injecting an extra memory matrix to assemble normal patterns during training. Based on this paradigm, the memory-based models have attracted attentions in recent years [15, 35, 44]. These methods always record the normal patterns into the memory, then recombine and re-weight the relevant patterns for reconstruction. However, if anomalous representations can be recovered through the re-weighting of normal patterns, the discriminating process may collapse. In contrast, our method compresses images into a discrete latent space, inspired by the vector quantization technology [21, 32], referring to ideas from lossy compression to relieve the model from modeling negligible information. Additionally, our hierarchical framework allows us to increase the size of the codebooks without incurring the codebook collapse problem, achieving meticulous anomaly detection at multiple levels with our prototype-oriented scoring method.

Table 1: Anomaly detection/localization results with AUROC metric on MVTec-AD. All methods are evaluated under the one-for-all settings. The learned model is applied to detect anomalies for all categories without fine-tuning. The best results are bold with black.

| | Category | US[46] | PSVDD[47] | PaDiM[48] | MKD[49] | DRAEM[50] | SimpleNet[51] | PatchCore[8] | RD4AD[52] | UTRAD[33] | UniAD[10] | Ours |
|---|---|---|---|---|---|---|---|---|---|---|---|---|
| Object | Bottle | 84.0 / 67.9 | 85.5 / 86.7 | 97.9 / 96.1 | 98.7 / 91.8 | 97.5 / 87.6 | 97.7 / 91.2 | **100** / 97.4 | 98.7 / 97.7 | **100** / 96.4 | 99.7 / 98.1 | **100** ± 0.00 / **98.3** ± 0.04 |
| | Cable | 60.0 / 78.3 | 64.4 / 62.2 | 70.9 / 81.0 | 78.2 / 89.3 | 57.8 / 71.3 | 87.6 / 88.1 | 95.3 / 93.6 | 85.0 / 83.1 | 97.8 / 97.1 | 95.2 / 97.3 | **99.0** ± 0.29 / **98.1** ± 0.04 |
| | Capsule | 57.6 / 85.5 | 61.3 / 83.1 | 73.4 / 96.9 | 68.3 / 88.3 | 65.3 / 50.5 | 78.3 / 89.7 | **96.8** / 98.0 | 95.5 / 98.5 | 82.0 / 97.2 | 86.9 / 98.5 | 95.4 ± 0.21 / **98.8** ± 0.01 |
| | Hazelnut | 95.8 / 93.7 | 83.9 / 97.4 | 85.5 / 96.3 | 97.1 / 91.2 | 93.7 / 96.9 | 99.2 / 95.7 | 99.3 / 97.6 | 87.1 / 98.7 | 99.8 / 98.2 | 99.8 / 98.1 | **100** ± 0.08 / **98.8** ± 0.02 |
| | Metal Nut | 62.7 / 76.6 | 80.9 / 96.0 | 88.0 / 84.8 | 64.9 / 64.2 | 72.8 / 62.2 | 85.1 / 90.9 | 99.1 / 96.3 | 99.4 / 94.1 | 94.7 / **96.4** | 99.2 / 94.8 | **99.9** ± 0.02 / 96.3 ± 0.09 |
| | Pill | 56.1 / 80.3 | 89.4 / 96.5 | 68.8 / 87.7 | 79.7 / 69.7 | 82.2 / 94.4 | 78.3 / 89.7 | 86.4 / 90.8 | 52.6 / 96.5 | 89.7 / 95.7 | 93.7 / 95.0 | **95.8** ± 0.49 / **97.1** ± 0.04 |
| | Screw | 66.9 / 90.8 | 80.9 / 74.3 | 56.9 / 94.1 | 75.6 / 92.1 | 92.0 / 95.5 | 45.5 / 93.7 | 94.2 / 98.9 | **97.3** / **99.4** | 75.1 / 95.2 | 87.5 / 98.3 | 95.6 ± 0.60 / 98.9 ± 0.05 |
| | Toothbrush | 57.8 / 86.9 | 99.4 / 98.0 | 95.3 / 95.6 | 75.3 / 88.9 | 90.6 / 97.7 | 94.7 / 97.5 | **100** / 98.8 | 99.4 / **99.0** | 89.7 / 97.5 | 94.2 / 98.4 | 93.6 ± 0.66 / 98.6 ± 0.04 |
| | Transistor | 61.0 / 68.3 | 77.5 / 78.5 | 86.6 / 92.3 | 73.4 / 71.7 | 74.8 / 64.5 | 82.0 / 86.0 | 98.9 / 92.3 | 92.4 / 86.4 | 92.0 / 91.5 | **99.8** / **97.9** | 99.7 ± 0.12 / **97.9** ± 0.04 |
| | Zipper | 78.6 / 84.2 | 77.8 / 95.1 | 79.7 / 94.8 | 87.4 / 86.1 | 98.8 / **98.3** | 99.1 / 97.0 | 97.1 / 95.7 | **99.6** / 98.1 | 95.5 / 97.3 | 95.8 / 96.8 | 97.9 ± 0.15 / 97.5 ± 0.10 |
| Texture | Carpet | 86.6 / 88.7 | 63.3 / 78.6 | 93.8 / 97.6 | 69.8 / 95.5 | 98.0 / 98.6 | 95.9 / 92.4 | 97.0 / 98.1 | 97.1 / **98.8** | 80.3 / 94.4 | 99.8 / 98.5 | **99.9** ± 0.03 / 98.7 ± 0.03 |
| | Grid | 69.2 / 64.5 | 66.0 / 70.8 | 73.9 / 71.0 | 83.8 / 82.3 | 99.3 / 98.7 | 49.8 / 46.7 | 91.4 / 98.4 | **99.7** / **99.2** | 93.9 / 95.2 | 98.2 / 96.5 | 97.0 ± 0.69 / 97.0 ± 0.06 |
| | Leather | 97.2 / 95.4 | 60.8 / 93.5 | 99.9 / 84.8 | 93.6 / 96.7 | 98.7 / 97.3 | 93.9 / 96.9 | **100** / 99.2 | **100** / **99.4** | 99.8 / 98.4 | **100** / 98.8 | **100** ± 0.00 / 98.8 ± 0.02 |
| | Tile | 93.7 / 82.7 | 88.3 / 92.1 | 93.3 / 80.5 | 89.5 / 85.3 | **99.8** / **98.0** | 93.7 / 93.1 | 96.0 / 90.3 | 97.5 / 95.6 | 98.8 / 94.2 | 99.3 / 91.8 | 99.2 ± 0.32 / 92.2 ± 0.42 |
| | Wood | 90.6 / 83.3 | 72.1 / 80.7 | 98.4 / 89.1 | 93.4 / 80.5 | **99.8** / **96.0** | 95.2 / 84.8 | 93.8 / 90.8 | 99.2 / 96.0 | 99.7 / 89.4 | 98.6 / 93.2 | 97.2 ± 0.40 / 92.4 ± 0.16 |
| | Mean | 74.5 / 81.8 | 76.8 / 85.6 | 84.2 / 89.5 | 81.9 / 84.9 | 88.1 / 87.2 | 85.1 / 88.9 | 96.4 / 95.7 | 93.4 / 96.0 | 92.6 / 95.6 | 96.5 / 96.8 | **98.0** ± 0.11 / **97.3** ± 0.05 |

Figure 4: Qualitative results for anomaly localization on MVTec-AD. 'Recon/Pred-Sing' and 'Recon/Pred-Hier' are reconstructions/score maps with single/hierarchical *VQLayer*, respectively.

# 4 Experiment

## 4.1 Datasets and Metrics

**MVTec-AD** [2] is a wildly-used industrial anomaly detection dataset with 15 classes, it covers more than 5k high-resolution images including objects and textures. For each class, the training samples are normal while the test samples can be either normal or anomalous. For each anomalous sample, the ground-truths of image label and segmentation are available for evaluation. In this paper, we investigate the unified case following [10], where only one model is used to handle all categories.

**VisA** [45] is a recently published large dataset, which consists of 9,621 normal and 1,200 anomalous high-resolution images. The dataset includes images with complex structures, objects placed in sporadic locations, and various types of objects. Anomalies encompass scratches, dents, color spots, cracks, and structural defects. All images are spatially resized to $224 \times 224$ to facilitate training.

**CIFAR-10** [45] is a classical image classification dataset of 10 categories. We adopt the challenging *many-versus-many* setting as in [10], where 5 classes are viewed as normal while the rest 5 classes are viewed as anomalies that remain unseen during training.

**Evaluation metrics:** We report the Area Under the Receiver Operator Curve (AUROC) on image-level anomaly detection and pixel-wise anomaly localization following the previous works [2, 10, 46].

## 4.2 Anomaly detection and localization performance on MVTec-AD

**Implementation details:** The input image size of MVTec-AD is $224 \times 224 \times 3$, after being fed into the pre-trained EfficientNet [20], the feature maps become $14 \times 14 \times 272$, namely, the patch size is 16. Then we reduce the channel dimension of each patch into 256, followed by feeding them into a 4-layer *vanTrans-enc* followed by the corresponding and a 4-layer *VQTrans-dec*. We use AdamW [53] with weight decay 0.0001 for optimization. Our model is trained for 1000 epochs on 2 GPUs (NVIDIA GeForce RTX 3080 10GB) with batch size 16. The learning rate is initialized as $1 \times 10^{-4}$ and dropped by 0.1 after 800 epochs. Our model is trained from scratch besides the pre-trained EfficientNet. For more details, please refer to the Appendix.

**Quantitative results of anomaly detection on MVTec-AD:** As shown in Table 1, the proposed HVQ-Trans generally outperforms all the competitive baselines. Specifically, our model surpasses PatchCore and UniAD by $1.6\%$ and $1.5\%$ on average. The former is a SOTA method under the one-for-one setting, the latter is a SOTA method under the one-for-all setting. Especially, in the one-for-all case, our model far exceeds UniAD by $8.5\%$ and $8.1\%$ on `Capsule` and `Screw`, respectively. We attribute this to that our model is more robust for identical shortcuts thanks to our well-designed

Table 2: Anomaly detection/location results (image AUROC, pixel AUROC) on VisA. Our model is applied to all categories without specific parameter-tuning on each category.

| Category | | DRAEM[50] | JNLD[55] | OmniAL[54] | UniAD[10] | Ours |
|---|---|---|---|---|---|---|
| Complex structure | PCB1 | 83.9 / 94.0 | 82.9 / 98.0 | 77.7 / 97.6 | 95.4 / 99.3 | **96.7 / 99.4** |
| | PCB2 | 81.7 / 94.1 | 79.1 / 95.0 | 81.0 / 93.9 | **93.6** / 97.8 | 93.4 / **98.0** |
| | PCB3 | 87.7 / 94.1 | 90.1 / 98.5 | 88.1 / 94.7 | 88.6 / **98.3** | 92.0 / **98.3** |
| | PCB4 | 87.1 / 72.3 | 96.2 / 97.5 | 95.3 / 97.1 | 99.4 / **97.9** | **99.5** / 97.7 |
| Multiple instances | Macaroni 1 | 68.6 / 89.8 | 90.5 / 93.3 | 92.6 / 98.6 | 92.2 / 99.3 | **93.1 / 99.4** |
| | Macaroni 2 | 60.3 / 83.2 | 71.3 / 92.1 | 75.2 / 97.9 | 85.9 / 98.0 | **86.2 / 98.5** |
| | Capsules | **89.6** / 96.6 | 91.4 / 99.6 | 90.6 / 99.4 | 72.0 / 98.3 | 77.1 / **99.0** |
| | Candles | 70.2 / 82.6 | 85.4 / 94.5 | 86.8 / 95.8 | 96.8 / 99.2 | **96.8 / 99.2** |
| Single instance | Cashew | 67.3 / 68.5 | 82.5 / 94.1 | 88.6 / 95.0 | 92.4 / 98.7 | **94.9 / 99.2** |
| | Chewing gum | 90.0 / 92.7 | 96.0 / 98.9 | 96.4 / 99.0 | 99.4 / **99.2** | **99.4** / 98.8 |
| | Fryum | 86.2 / 83.2 | 91.9 / 90.0 | 94.6 / 92.1 | 89.8 / 97.7 | 90.4 / 97.7 |
| | Pipe fryum | 87.1 / 72.3 | 87.5 / 92.5 | 86.1 / 98.2 | 97.4 / 99.2 | **98.5 / 99.4** |
| Mean | | 80.5 / 87.0 | 87.1 / 95.2 | 87.8 / 96.6 | 91.9 / 98.6 | **93.2 / 98.7** |

architecture and algorithm. Notably, the performance of various categories are significantly different (Toothbrush), which might due to the data distribution of each category is different, corresponding to different requests for representation ability. The switching mechanism learns individual codebooks and experts for each class, decreasing the distortion between multiple classes. To sum up, our model is proven to be effective and efficient for one-for-all anomaly detection applications.

**Quantitative results of anomaly localization on MVTec-AD:** Anomaly localization aims to detect the anomalous regions given anomalous samples. The localization results under the one-for-all setting are shown in Table 1. As we can see, our model outperforms all the competitive baselines on average. Specifically, as a strong SOTA baseline, UniAD is also left behind by our model by 0.5% on both settings averagely speaking. We attribute this to that the hierarchical VQ also plays an important role in precise localization besides the learnable query embeddings verified by UniAD since our HVQ-Trans only employs the traditional query embeddings. Moreover, localization requires more precise position information compared with its detection counterpart, a proper measurement alignment to enhance the real anomalous regions may be important, which is exactly what our POT is good at. Another interesting finding is reported in Appendix that although there exists information loss in reconstructing normal images, such information loss is even more significant in reconstructing anomalous images. Hence, the actual anomaly localization performance is improved.

**Qualitative results of anomaly localization on MVTec-AD:** As shown in Fig. 4, our method can successfully recover the anomalous regions with their corresponding normal patterns for both object anomalies (Left) and texture damages (Right). It can be seen that the model with hierarchical VQ layers could better generate normal patterns at abnormal regions, resulting in more accurate anomaly localization. More qualitative results are given in Appendix.

### 4.3 Anomaly detection on VisA

**Quantitative results on VisA:** Compared to MVTecAD, VisA poses greater difficulty due to its more complex structures and scenes with multiple misaligned instances. Table 2 demonstrates the superior performance of our model in comparison to the other three reconstruction-based methods under the unified setting. Our proposed model surpasses the best of the comparison methods, *i.e.*, UniAD, by 1.3% on image-AUROC, leading to significantly superior performances than the modest recent unified model OmniAL [54] (5.4% and 2.1% on detection and localization).

**Qualitative results on VisA:** Figure 5 illustrates the impressive performance of our reconstruction and localization in various categories. Even in multi-instance scenes, Our model effectively restores the anomaly region to its normal state. The reconstructed images exhibit a high level of fidelity, closely matching the appearance of normal regions and meeting expectations in recovering anomalies. More qualitative results are given in Appendix.

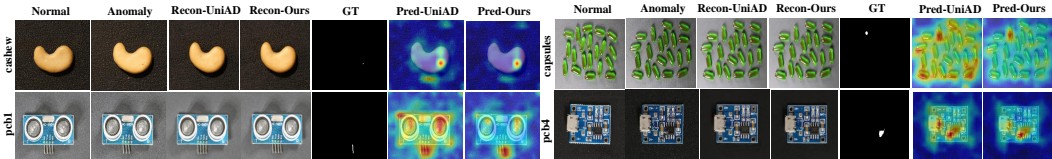

Figure 5: Qualitative results for anomaly localization on VisA.

Table 3: Anomaly detection results with AUROC metric on CIFAR-10 under the one-for-all setting. Normal indices indicate the names of normal classes. The best results are bold with black.

| Normal Indices | US[46] | FCDD[56] | FCDD+OE[56] | PANDA[57] | MKD[49] | UniAD[10] | Ours |
|---|---|---|---|---|---|---|---|
| {01234} | 51.3 | 55.0 | 71.8 | 66.6 | 64.2 | **84.4** | 82.6 ± 0.02 |
| {56789} | 51.3 | 50.3 | 73.7 | 73.2 | 69.3 | 80.9 | **84.3** ± 0.03 |
| {02468} | 63.9 | 59.2 | 85.3 | 77.1 | 76.4 | **93.0** | 92.4 ± 0.10 |
| {13579} | 56.8 | 58.5 | 85.0 | 72.9 | 78.7 | 90.6 | **91.9** ± 0.06 |
| Mean | 55.9 | 55.8 | 78.9 | 72.4 | 72.1 | 87.2 | **87.8** ± 0.04 |

Table 4: Ablation studies with AUROC metric on MVTec-AD. w/o VQ means without VQ.

| w/o VQ | with VQ | | Switching | | POT | Detection | Localization |
|---|---|---|---|---|---|---|---|
| | Single | Hierarchical | Codebook-Switching | Expert-Switching | | | |
| ✓ | - | - | - | - | - | 70.5 | 81.4 |
| - | ✓ | - | - | - | - | 96.2 | 96.8 |
| - | ✓ | - | ✓ | - | - | 96.4 | 96.8 |
| - | - | ✓ | - | - | - | 97.1 | 96.9 |
| - | - | ✓ | ✓ | - | - | 97.2 | 97.0 |
| - | - | ✓ | - | - | ✓ | 97.4 | 97.2 |
| - | - | ✓ | ✓ | ✓ | - | 97.6 | 97.2 |
| - | - | ✓ | ✓ | ✓ | ✓ | **98.0** | **97.3** |

## 4.4 Anomaly detection on CIFAR-10

**Implementation details:** In order to implement *many-versus-many* anomaly detection, we select 5 normal classes while the rest classes are viewed as anomalies. As shown in Table 3, {01234} means the normal samples include images from classes 0, 1, 2, 3, and 4, while the images from 5, 6, 7, 8, and 9 are anomalies. For statistical robustness, we repeat the splitting and obtain four combinations.

**Quantitative results on CIFAR-10:** As shown in Table 3, the performance of our model surpasses all the other competitors under *many-versus-many* setting with each dataset splitting. Besides, CIFAR-10 dataset itself is more complex than MVTec-AD because of its poor shooting conditions. Hence, the one-for-all setting on CIFAR-10 imposes stricter requirements for the model to exactly distinguish normal patterns from anomalous interference. Therefore, the substantial improvement further verifies the superiority of our method.

## 4.5 Ablation studies

**Component study:** To verify the effectiveness of the proposed modules, including single *VQLayer*, hierarchical *VQLayers*, switching mechanism for codebooks and reconstruction experts, and POT scoring, we implement extensive ablation studies on MVTec-AD. As shown in Table 4, we have the following observations: (i) The performance of the model without VQ drops by nearly 26% (96.4 to 70.5), which demonstrates that VQ plays the key role in anomaly detection and the vanilla Transformer is powerful to well reconstruct both the normal and anomaly. Our VQ module acts as the information bottleneck where only the normal information is allowed to pass through, thus degrading the reconstruction of anomalous; (ii) The hierarchical structure also presents performance gain since it provides local access to multi-level codebooks, thus reducing the search complexity per layer and releasing the codebook collapse issue; (iii) The switching brings slight improvements on MVTec-AD, while it achieves significant gains up to 3.2% on CIFAR-10, as shown in the Appendix. We attribute this to the different degrees of difficulty posed by the two dataset distributions. Our switching mechanism plays a more critical role for the complex datasets, *i.e.*, CIFAR-10 in this case; (iv) The POT module is effective in detecting and localizing anomalies due to its cascade measurement alignment property.

Table 5: Different hierarchical structures.

| Structure | Detection | Localization |
|---|---|---|
| $h^l \rightarrow z^l$ | 94.9 | 96.5 |
| $h^{l-1} \oplus h^l \rightarrow z^l$ | 97.2 | 97.0 |
| $h^{l-1} \oplus \theta \rightarrow z^l$ | **98.0** | **97.3** |

Table 6: Different prototype numbers $K$.

| $K$ | Detection | Localization |
|---|---|---|
| 1024 | 97.1 | 97.2 |
| 512 | **98.0** | **97.3** |
| 256 | 97.0 | 97.1 |

**Different hierarchies:** We demonstrate experiments to investigate the impact of various hierarchies, as shown in Table 5, where the different information is encoded in different layers. While the multi-level features $h^l$ are concatenated with the global prototypes $\theta$, the joint performance over anomaly detection and localization increases. One possible explanation is that the global prototypes $\theta$ at much higher abstraction levels may result in efficient latent representation for anomaly detection.

**Robustness to prototype number:** We conduct the experiments by using different prototype numbers $K$ and show the AUC values in Table 6. Given different numbers of prototypes, our HVQ-Trans can still surpass most of the competitors in Table 1, proving the robustness of our method.

**Visualizing how the prototypes works:** In order to investigate what exactly the prototypes have learned, we further train a mapping function from the reconstructed feature to the observation space and present the visualization results in Fig. 6. We compulsively instructed the model to reconstruct the patches within the red box using prototypes from the codebook of a different, irrelevant category. As demonstrated in the figure, the reconstructed region is highly correlated to the prototypes, *e.g.*. the centering region in the 'Cable' image is reconstructed as 'Grid'. The observations confirm that our iconic prototypes accurately represent typical normal patterns of each category, and only reconstruct the corresponding normal appearances as intended.

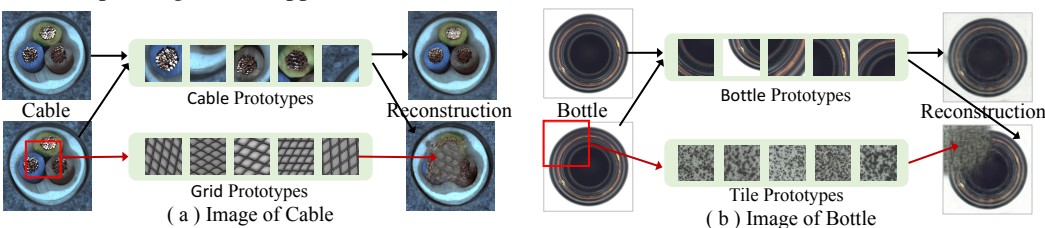

Figure 6: The top row illustrates the images can be well reconstructed with switched prototypes. The bottom row displays when the patches in the red box are forced to be reconstructed with irrelevant prototypes, the reconstructed region is in accordance with the given prototypes.

## 5   Conclusion

We introduce a unified model, HVQ-Trans, for multi-class Unsupervised Anomaly Detection under the one-for-all setting. The latent space is modeled as hierarchical discrete prototypes learned from normal training data. We vector quantize visual features to reconstruct normal patterns and employ a switching mechanism for codebook selection and exquisite reconstruction. Our hierarchical designation incorporates multi-level normative information and encourages the model to reconstruct anomalous images as normal. Furthermore, we propose the hierarchical prototype-oriented optimal transport module to regulate the prototypes and calibrate the anomaly score. Under the one-for-all setting, our model significantly surpasses competitors on MVTec-AD and VisA datasets, and provides visualization and interpretability for both anomaly localization and detection.

**Discussion:** In this work, the category labels are assumed to be available during the training stage. How to incorporate the model with clustering methods rather than category labels should be further studied. In practice, our model can assemble the normal iconic prototypes which may facilitate domain adaption for real scenes, and be potentially applied to time series, text, and video data. However, anomaly detection for video surveillance or social multimedia may raise privacy concerns.

## Acknowledgements

This work was supported in part by the National Natural Science Foundation of China (NSFC) under Grant U21B2006; in part by the NSFC under Grant 82172860; in part by the NSFC under Grant 6220010437; in part by the Shaanxi Youth Innovation Team Project; in part by the Fundamental Research Funds for the Central Universities QTZX23037 and QTZX22160; in part by the Fundation of Aerospace under Grant SAST2021012; in part by the 111 Project under Grant B18039.

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
