# Appendix of "Hierarchical Vector Quantized Transformer for Multi-class Unsupervised Anomaly Detection"

## 1  Supplementary Implementation Details:

**MVTec-AD:**   The input image size of MVTec-AD is $224 \times 224 \times 3$, after being fed into the pre-trained EfficientNet [1], the feature maps become $14 \times 14 \times 272$, namely, the patch size is 16. Our model is trained for 1000 epochs on 2 GPUs (NVIDIA GeForce RTX 3080 10GB) with batch size 16. The hyperparameters $\beta$ and $\alpha$ are set as 0.5 and 0.01 for each layer.

**VisA:**   The input image size of VisA is resized to $224 \times 224 \times 3$ and the network architectures and hyperparameters are same as MVTec-AD.

**CIFAR-10:**   The image size is set to 224 x 224, and the feature size is 14 x 14. Since the anomalies in CIFAR-10 are semantically dissimilar objects, features in deeper layers, which contain more semantic information, are expected to be more useful. Therefore, we selected the feature maps from stages 1 to 5 and resized and concatenated them to form a 720-channel feature map. The channel dimension was reduced to 256. Our model was trained for 1000 epochs on 2 GPUs (NVIDIA GeForce RTX 3080 10GB) with a batch size of 64, using the AdamW optimizer [2] with a weight decay of $1 \times 10^{-4}$. The initial learning rate was set to $1 \times 10^{-4}$ and was reduced by a factor of 0.1 after 800 epochs. The encoder and decoder layers were both set to 4. The hyperparameters $\beta$ and $\alpha$ are set to 0.5 and 0.01 for each layer.

## 2  More Details of Optimization:

Note that there is no real gradient of the $argmax$ operation in the vector quantization layer, however, we approximate the gradient by copying gradients from the refined visual tokens $z^l$ to the visual tokens $h^l$ for $l = 0, ..., L$. Although one could use the subgradient through the quantization operation, we opted for a simpler estimator for the initial experiments in this paper, which worked well. During forward computation, the nearest prototype is passed to the decoder. During the backward pass, the gradient is passed unaltered to the encoder. As the output representation of the encoder and the input to the decoder share the same D-dimensional space, the gradients contain valuable information about how the encoder needs to modify its output to reduce the reconstruction loss.

Overall, the proposed HVQ-Trans incorporates five terms into its objective, specified as:

$$
\mathbb{L}_{HVQ-Trans} = ||\boldsymbol{h}^0 - \tilde{\boldsymbol{h}}^0||_2^2 + \sum_{l=1}^{L} \left[ ||\mathrm{sg}(\boldsymbol{h}^l) - \boldsymbol{e}^l||_2^2 + \beta^l||\boldsymbol{h}^l - \mathrm{sg}(\boldsymbol{e}^l)||_2^2 + \alpha^l \mathbb{L}_{POT}^l \right]
$$
$$
- \sum_{j=1}^{M} p_j(\boldsymbol{x}) \log \mathcal{P}_x,
$$

(1)

where the $\mathrm{sg}(\cdot)$ refers to the stop-gradient operation and $\mathcal{P}_x$ the category label. Both $\beta^l$ and $\alpha^l$ are hyperparameters, however, we found that the resulting algorithm is quite resilient to variations in their values.

37th Conference on Neural Information Processing Systems (NeurIPS 2023).

(1) The first term refers to the reconstruction loss which optimizes the decoder and the encoder. The ELBO of our variational autoencoder should include both a reconstruction likelihood and a KL term. Since we assume a uniform prior for prototypes, the KL term that usually appears in the Evidence Lower Bound (ELBO) is constant, w.r.t. the KL divergence can thus be ignored for training.

To be specific, our proposed model can be viewed as the hierarchical discrete variational autoencoding, the posterior of which is defined as the categorical distribution below:

$$q(\boldsymbol{z}^l|\boldsymbol{x}) = \begin{cases} 1, \text{for } k = \arg\min_{j} ||h^l - e^l_j||, \\ 0, \text{otherwise,} \end{cases} \tag{2}$$

Note the posterior distribution $q(z^l = k|x)$ is deterministic, and we define the simple uniform prior over hierarchical discrete prototypes. Thus, the KL divergence in ELBO of our proposed model can be computed as a constant $Llog K$, thus being ignored for training.

(2) The second one in the scope of summation along $L$ layers is the hierarchical prototypical loss, pushing the selected prototype $\boldsymbol{e}^l$ closer to the visual token $\boldsymbol{h}^l$. This loss uses the $l2$ error to move the prototypes toward the encoder outputs, which is only used for updating the dictionary. Following [3], we use the exponential moving average updates for codebooks.

(3) The third term denotes the hierarchical commitment loss, optimizing the encoder by encouraging the output of the encoder $\boldsymbol{h}^l$ to stay close to the chosen prototype and prevent it from fluctuating too frequently from one prototype to another. The commitment loss ensures that the encoder commits to an embedding and its output does not grow.

(4) The fourth term is the POT loss, which helps the model learn the prototypes of the normal codebook at each certain layer.

(5) The last term is the cross entropy loss for training the classifier to adaptively switch the proper reconstruction expert and codebook.

## 3 Ablation studies on CIFAR-10:

To evaluate the effectiveness of our proposed modules, including the single *VQLayer*, hierarchical *VQLayers*, codebook-switching and reconstruction expert-switching mechanisms, and the POT scoring method, we conducted extensive ablation studies on CIFAR-10. As shown in Table 1, we made the following observations:

(i) The codebook-switching mechanism plays an important role in anomaly detection on CIFAR-10, resulting in significant performance gains of up to 3.2%. This might due to the images in CIFAR-10 are more complex, causing the prototypes of different categories to be easily confused without our codebook-switching mechanism. Consequently, the reconstruction error becomes confounding around each category, resulting in anomalies being more difficult to recognize.

(ii) The hierarchical structure also demonstrated performance gains since it provides local access to multi-level codebooks, thereby reducing the search complexity per layer and releasing the codebook collapse issue.

(iii) The expert-switching mechanism also leads to performance improvements on CIFAR-10 by accurately reconstructing features for each category.

(iv) The POT module effectively detected and localized anomalies due to its well-defined measurement alignment property.

Finally, the overall model that incorporates all of these components achieved the best results, demonstrating the overall effectiveness of our framework.

## 4 The Reconstruction Error:

To evaluate the reconstruction ability of our model for normal and anomalous images, we computed the average reconstruction loss $||\boldsymbol{h}^0 - \tilde{\boldsymbol{h}}^0||_2^2$ for each category in MVTec-AD. As depicted in Fig. 1, the normal images result in lower reconstruction error, while the anomalous images exhibit worse

Table 1: Ablation studies with AUROC metric on CIFAR-10. w/o VQ means without VQ.

| with VQ | | Switching | | POT | Detection |
|---|---|---|---|---|---|
| Single | Hierarchical | Codebook-Switching | Expert-Switching | | |
| ✓ | - | - | - | - | 71.1 |
| ✓ | - | ✓ | - | - | 74.3 |
| - | ✓ | ✓ | - | - | 77.6 |
| - | ✓ | ✓ | ✓ | - | 81.6 |
| - | ✓ | ✓ | ✓ | ✓ | **87.8** |

reconstruction results. Our model consistently shows obvious reconstruction discrepancies between normal and abnormal images. It is worth noting that we computed the error by averaging the feature loss of the entire image, with the abnormal region occupying only a limited partition. Despite this, the discrepancy is evident at the image level, indicating that the reconstruction errors are even more pronounced in the abnormal regions. This demonstrates that while there may be some information loss in vector quantization, such loss is more pronounced in reconstructing anomalous regions compared to normal regions.

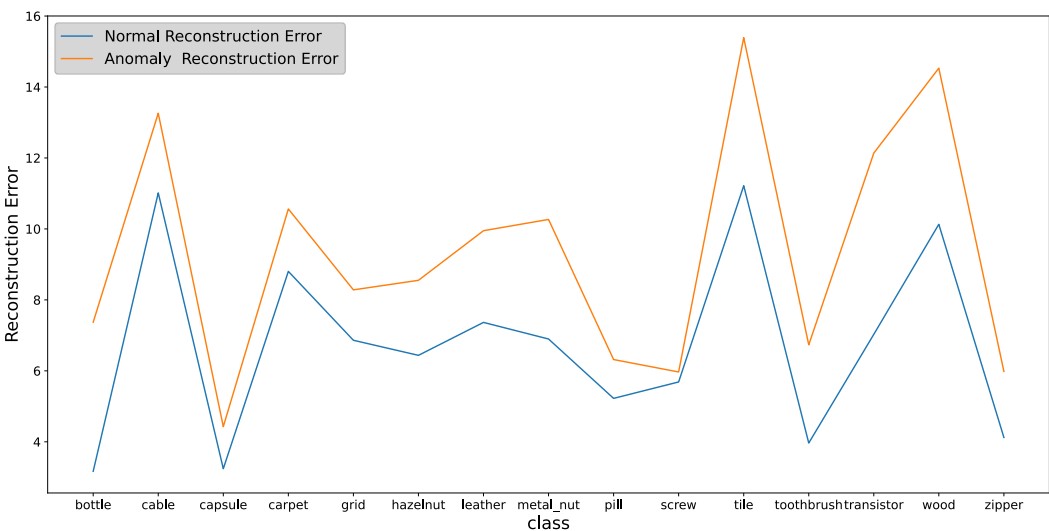

Figure 1: The average reconstruction error of normal and anomalous images of different categories in MVTec-AD.

## 5 More Qualitative Results of Anomaly Localization:

The qualitative results for anomaly localization on MVTec-AD are shown in Fig. 2. All 15 categories can be handled by our HVQ-Trans. For various kinds of anomalies, the anomalous regions can be reconstructed as the normal patterns, then can be accurately localized by reconstruction differences. These qualitative results are not apple-picking, which can demonstrate the effectiveness of our HVQ-Trans.

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

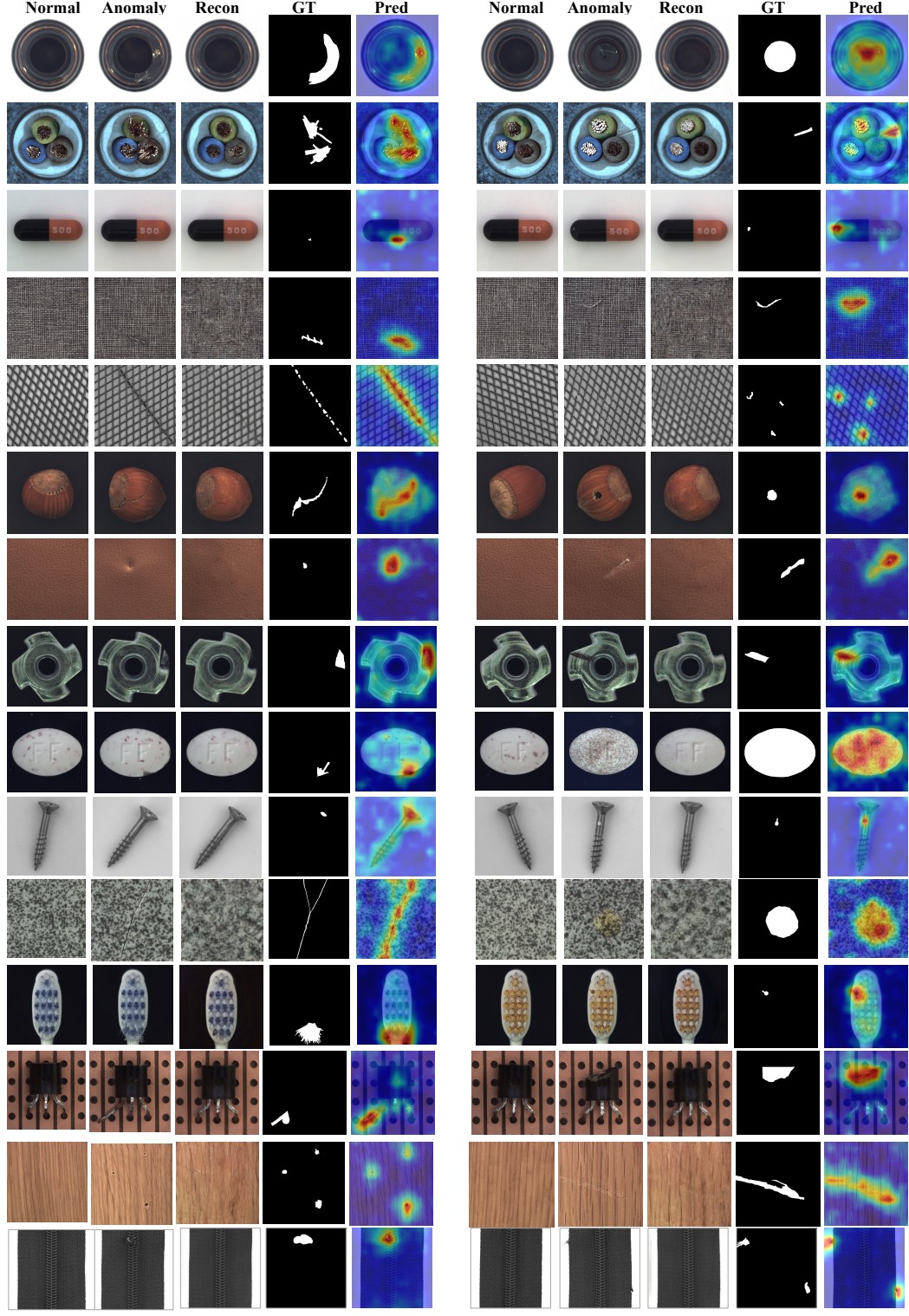

Figure 2: Qualitative visualizations of 15 categories of MVTec-AD. Qualitative results for anomaly localization on MVTec-AD under the one-for-all (unified) case. From left to right: normal sample as the reference, anomaly, our reconstruction, ground-truth, and our predicted anomaly map.