# OpenReview forum: "Hierarchical Vector Quantized Transformer for Multi-class Unsupervised Anomaly Detection"
_NeurIPS.cc/2023/Conference — NeurIPS 2023 poster_

### Official Review · Reviewer_uU2U · 2023-07-04

**Soundness:** 3 good
**Presentation:** 3 good
**Contribution:** 2 fair
**Rating:** 5
**Confidence:** 5

**Summary:**

This work builds a unified framework for multi-class anomaly detection (AD) by using normal images only. It identifies the identical shortcut issue of the reconstruction-based AD methods and tries to alleviate it by augmenting the memory with hierarchical discrete iconic prototypes. A switching mechanism is employed to deal with multi-class scenario. Furthermore, the hierarchical prototype-oriented optimal transport module is used to calibrate the anomaly scores. Experiments on MvTec-AD and CIFAR-10 demonstrate the superiority of the method.

**Strengths:**

1. The motivation of using discrete iconic prototypes makes sense to alleviate the “shortcut” of reconstruction on abnormal images.
2. It achieves state-of-the-art performance on the two popular benchmarks.
3. The paper is easy-to-follow and well-written with clear motivation for each module.

**Weaknesses:**

1. Missing necessary comparison on additional datasets, e.g., the VisA dataset [1], a multi-class industrial anomaly detection dataset.
2. As a core contribution, the details of vector quantization (VQ) (e.g., the quantization method) are not thoroughly presented and some related questions are not discussed, e.g., how different quantization precisions/methods affect the performance?

[1] Zou, Yang, Jongheon Jeong, Latha Pemula, Dongqing Zhang, and Onkar Dabeer. "Spot-the-difference self-supervised pre-training for anomaly detection and segmentation." In European Conference on Computer Vision, 2022.

**Questions:**

1. In Table 1, why the results under one-for-all setting are better than that under one-for-one setting? For example, the results on Capsule is far lower under one-for-one setting. Though the authors claim that the increased data diversity is beneficial, it requires more experiments to support the argument, e.g., try taking images from different N (N >= 2) classes for training.
2. The ablation study is incomplete. Missing the results when only Hierarchical VQ is enabled.

**Limitations:**

Limitations are discussed in the Discussion section and potential negative societal impacts are not mentioned.

---

> ### Author Rebuttal · Authors · 2023-08-10
>
> Thanks a lot for your effort in reviewing!
>
> A1: According to the reviewer's suggestion, we have implemented experiments on the VisA dataset. The performance is shown in Table 4 of the unloaded PDF. As we can see, our model surpasses the previous one-for-all SOTA method, UniAD, by 1.3\% and 0.1\% for anomaly detection and localization, respectively.
> Moreover, our model surpasses the previous one-for-one SOTA method, DRAEM, by 12.7\% and 11.7\% for anomaly detection and localization. This demonstrates the effectiveness and robustness on VisA dataset, which is more challenging than the MVTec-AD dataset.
>
> We will add the results on the VisA dataset to the revised manuscript.
>
> A2: The hierarchical VQ-based layers layer-wisely quantize the visual tokens $h^l$ to the prototypes $e_k^l$ in the learnable codebooks $E^l \in R^{K \times C}$.
> For the output $h^L$ final layer of encoder, we replace the visual tokens $h^L$ with its most similar prototypes $e_i^L$ in the codebook $E^L$ as: $ \theta = Quantize(h^L)=e_i^L,  i= {min}_{j} \| h^L - e^L_j\|_2^2,$ where $\theta$ represents the global quantized vector. The visual token is quantized based on its distance to the prototype vectors in the codebook $E^L$, such that each visual token is replaced by the nearest prototype vector in the codebook, and is transmitted to the decoder.
>
> Moreover, we find that merging fine-grained concrete information with abstraction-level semantics is critical for robust anomaly detection. Hence, we fuse the multi-level visual tokens with the global quantized vector $\theta$ to learn hierarchical prototypes, maximizing the preserved nominal information, stated as equation 1 in the main paper. Intuitively, we hierarchically replace visual tokens $h^l$ with their most similar prototypes in the codebook $E^l$ as quantized vector $z^l$.
>
> Note that there is no real gradient defined for quantization, however we approximate the gradient similar to the straight-through estimator and just copy gradients from decoder input $z^l$ to embedding before quantization. One could also use the subgradient through the quantisation operation, but this simple estimator worked well for the initial experiments in this paper.
>
> We promise to add detailed and clear descriptions of vector quantization in the revised paper.
>
> \textbf{Affection of VQ:} As shown in Table 4, the vanilla Transformer obtain 70.5 and 81.4 on AD and AL, while increasing to 96.4 and 96.8 by adding the VQ-layers.
> The performance of the model increases by nearly 26\%, which demonstrates that VQ plays the key role in anomaly detection. Our VQ module acts as the information bottleneck where only the normal information is allowed to pass through, leading to larger feature migration and information loss for anomalies.
> This discrepancy of information loss for normal and anomaly serves as a key factor in effective anomaly detection.
>
> A1-to-Q1: We think there are two main reasons causing the better performance under the one-for-all setting:
>
> 1) The amount of training data is bigger and the training data diversity is increased. Thus, the model representation ability can be improved. Furthermore, diverse training data force the model to learn robust and discriminative representations and separate from each category, leading to tighter boundaries modeled by prototypes of each category.
> We have replenished experiment on smaller $N$ in Table 5 of the uploaded file. Specifically, we randomly choose 5 categories from all the 15 categories of MVTec-AD for 3 times without overlap, namely $N=5$ in this case. We find that the mean AUCs of detection and localization at this moment are 97.6\% and 97.3\%, respectively. Compared with our model with $N=15$, the detection performance drops 0.4\%, thus it could verify the benefits of training data diversity to some extent.
>
> 2) The switching mechanism under the one-for-all setting will classify each input image into a single category and choose the corresponding codebook and expert for reconstruction. For the normal images, it is highly likely to be classified into the correct category and thus switch the proper reconstruction expert and codebook. For the abnormal images, there remains big uncertainty that which reconstruction expert and codebook will be switched, because the anomalies are unseen during training. Thus, the reconstruction uncertainty of the abnormal image is increased. Noting that the difference between normal and abnormal is the key factor deciding the anomaly detection performance. Thus, this uncertainty of anomalous could improve the performance under the one-for-all setting, which also proves that the switching mechanism of our proposed method is more suitable for multi-class anomaly detection.
>
> 3) The data distribution of each category is different, corresponding to different requests for model representation ability. It's worth noting that our network architecture is simply implemented without many tricks, such as hyperparameter tuning for each category (\eg. prototype numbers, latent dimension) or adaptive architecture for individual categories. After tuning parameters on the Capsule category, we improve the performance from 88.3 to 94.7 on this category. However, the original intention
> of the one-for-all case is to save computational resources and achieve efficient modeling for all the category simultaneously.
> On the whole, our model can efficiently get decent performances under both the one-for-all and one-for-one settings.
>
> A2-to-Q2: We have replenished extra ablation studies in Fig. 5 of the uploaded file. As we can see that only employing Hierarchical VQ achieves 97.1 \% and 96.9 \% in detection and location, achieving 26.6\% and 15.5\% compared with the vanilla Transformer. The results can verify that the main performance gain come from our proposed hierarchical VQ-based framework.
>
> Potential negative societal impacts: Anomaly detection for video surveillance or social multimedia may raise privacy concerns.

---

> > ### Comment · Reviewer_uU2U · 2023-08-20
> > **Keep my rating**
> >
> > I appreciate that the authors provide additional experiments (main results on the VisA dataset and the ablation study) and detailed descriptions on the vector quantization. After carefully investigating other reviewers' comments and the rebuttal, the authors solve most of my concerns; however, I would like to keep my original rating due to the limited technical novelty of quantization on memory mechanism, which has been widely explored in the literature.

---

> > > ### Author Response · Authors · 2023-08-21
> > >
> > > Thank you a lot for your effort in reviewing this submission!
> > >
> > > Please allow us to explain the difference between our model and the previous memory-based works.
> > >
> > > **The ''short cut'' problem-oriented motivation:** We aim to model discrete space to intrinsically prevent anomaly information leakage into reconstruction. In contrast, the existing memory-based methods recombine and aggregate the discrete memory items, falling into an unknown continuous latent space which might be distorted. In contrast, we force the anomaly features to be replaced by a single discrete prototype. In addition, we would like to highlight that simplicity is the ultimate form of sophistication. Therefore, instead of saying the novelty reside in quantization technique, what we want to express is that VQ is a proper pathway to optimize prototypes for crimping information bottleneck, which is an effective way to achieve our purpose rather than the novelty itself.
> > >
> > > **Hierarchical designation is necessary and crucial:**
> > > Rather than directly employ the original vector quantization mechanism, we elaborately investigate a cascaded VQ transformer to overcome the **"prototype collapse"** problem: At some point during training, a part of latent codes in the codebook may no longer work and the modeling capacity is limited by the discrete representations, resulting in collapsed reconstruction.
> > > **The hierarchical designation is not easy**, as it's sophisticated to balance the prototypes of different levels  to maximize the nominal information available. As shown in table 5, another hierarchical structure results in large performance drop. Furthermore, the hierarchical designation fits for the hierarchical nature of vision, and matches to calibration of the anomaly score with hierarchical prototype-oriented optimal transport, which could also reduce the decoding search time and retain high inference speeds.  Even though HVQ-Trans is based on vector quantization mechanism, it achieves significantly better anomaly detection performance than the vanilla VQ model and the previous memory-base algorithms.
> > >
> > > **VQ-based Transformer:**
> > > This paper proposed a original way to leverage the iconic prototypes into Transformer, which properly fuses the hierarchical nominal informations and tighten the information bottleneck. We believe there is a crytic tradeoff for the information bottleneck, where more information passby will cause the anomaly leakage and less information passby will lead to poor reconstruction. Our HVQ-Trans could well handle this connotative issue.
> > >
> > > **Switching prototypes rightly fits for one-for-all setting:** Our model targets for the challenging 'one-for-all' case, which suffers from the 'identical shortcut' issue more severely, as the model generalizability increases due to the complex data distribution of multiple classes. To tightly fit the unified case, we propose the switching prototypes to set corresponding codebook for different data distribution, thus alleviating the `identical shortcut' issue. Please also consider that the paper also provides novel techniques, such as switching experts and the prototype-oriented learning and scoring, to fit for the challenging and practical one-for-all setting.
> > >
> > >
> > > We want to express the quantization technique is **not just simply employed in this paper as explored in the literature.** In the context of visual anomaly detection, we firmly believe that our method provide a new way to model discrete space without confusion, which intrinsically prevents anomaly information leakage into reconstruction, constitutes meaningful contributions.
> > >
> > > As the deadline for discussions between reviewers and authors is approaching, we sincerely invite you to take a moment from your busy schedule to read our reply. Thanks again for your valuable time! We are genuinely grateful for your response and help.
> > >
> > > Best regards,
> > >
> > > Anonymous author

---

### Official Review · Reviewer_ksQL · 2023-07-05

**Soundness:** 1 poor
**Presentation:** 2 fair
**Contribution:** 1 poor
**Rating:** 3
**Confidence:** 4

**Summary:**

This paper proposes a variational autoencoding framework for unsupervised anomaly detection. This work addresses the identical shortcut issue by preserving the typical normal patterns as discrete iconic prototypes and also overcomes the problems of prototype collapse problem.

**Strengths:**

It designs a network from the perspective of solving an issue identical shortcut. The method achieves good experimental results on classical datasets.

**Weaknesses:**

1. The descriptions of the figures need to be clearer. For example, in Fig.1, it would be helpful to label each image and coordinate system to indicate what they represent. Additionally, labeling the subfigures with (a) and (b) would make the explanation clearer, such as in Fig.5.
2. The performance on chosen datasets are close to saturation, especially in some categories where the results achieve 100, this is unreasonable. In this field, more challenging datasets should be tested to further push the boundaries of research.
3. The performance of proposed method under one-for-one setting still has a gap compared to the SOTA method.
4. The innovation is limited, where most of the ideas have already be proposed.


**Questions:**

Why the proposed methods under one-for-one setting has lower performance than the one-for-all setting, which differs from those in most other methods? What is the reason for the significant performance differences of the proposed method on different categories, such as poor performance on “Toothbrush”, and is this explainable?

**Limitations:**

The authors have clarified the limitation of this method that the category labels are assumed to be available during the training stage， and identify this issue as future study by incorporating the model with clustering methods.

---

> ### Author Rebuttal · Authors · 2023-08-10
>
> A1: Thanks for your suggestions. We have revised the descriptions and figures to be clearer. Due to the page limitation, we show revised Fig. 1 and Fig. 5 in the uploaded PDF. We promise to carefully polish all the descriptions and figures in our revised version.
>
> A2: Thanks for your suggestion.
>
> 1) As the reviewer mentioned, the performance on the wildly-utilized industrial anomaly detection dataset MVTec-AD tends to saturation. This might be because the images in the dataset have simple backgrounds and high resolution, thus the anomalies are relatively easy to be detected. In contrast, our model targets for the challenging ‘one-for-all’ case, which
> suffers from the ‘identical shortcut’ issue more severely, as the model generalizability increases due to the complex data distribution of multiple classes. The performance under the ‘one-for-all’ setting on MVTec-AD still needs to be improved.
>
> 2) It's worth noting that we also demonstrate experiments on a challenging dataset, \ie, CIFAR-10, which contains complex background and various anomalies.
> Our model achieves 86.1, while the comparison models result in \{55.9, 55.8, 78.9, 72.4, 72.1, 82.1\} for anomaly detection.
>
> 3) According to the reviewer's suggestion, we have further implemented experiments on another industrial anomaly detection dataset, the VisA dataset. The performance is shown in Table 4 of the unloaded PDF. As we can see, our model surpasses the previous SOTA method, UniAD, by 1.3\% for anomaly detection on the new dataset. We will add the results on the VisA dataset to the revised manuscript.
>
> A3: We hope to address this concern from the following aspects:
>
> 1. Focusing on the one-for-all model handling the one-for-one task, our model surpasses UniAD (the SOTA method for unified case) from 96.6 to 96.9 for anomaly detection, and improves the anomaly localization performance from 96.6 to 97.1. Thus, under this fair comparison setting, our model could better generalize to the one-for-one setting.
>
> 2. Without exhausted individual parameter tuning for each category, we set the same hyperparameters for each category, which might lead to a gap. For example, the category of texture or simple objects, such as bottle and grid, possess limited iconic normal patterns. The corresponding number of prototypes might be smaller than those complex objects. Rather than specific and exhausted model tuning for each category, our network architecture is simply implemented under the one-for-one setting. Accordingly, we particularly tune the hyperparameters on the 'Screw' category in the MVTec-AD dataset, and result in 94.5\% performance, surpassing all the comparison methods.
> Besides, although DRAEM (the SOTA one-for-one method) achieves good performances under the separate case, it drops severely (9.9\% and 10.1\%) changing to the unified case. Thus, on the whole, our model achieves decent performance in both settings.
>
> 3. Last but not least, we want to reclaim that our purpose is to alleviate the pivotal ‘identical shortcut’ issue in anomaly detection under the challenging one-for-all setting. When changing the one-for-one setting to the one-for-all setting, most of the existing alternatives fail to handle the challenging cases. Apart from the UniAD getting a slight drop (0.1\%) for detection, the other methods drop severally (12.4\% on average) when changing to the one-for-all setting.
> However, our model gets a 1.1\% performance gain for anomaly detection. This indicates the framework of UniAD and our model could handle the challenging setting. Thus, instead of saying there is a gap compared to the SOTA method on the separate case, what we want to express is that there is no performance drop (even a slight improvement) from the separate case to the unified case.
>
> A4: We think there are two main reasons causing the better performance under the one-for-all setting:
>
> 1) The amount of training data is bigger and the training data diversity is increased. Thus, the model representation ability can be improved. Furthermore, diverse training data force the model to learn robust and discriminative representations and separate from each category, leading to tighter boundaries modeled by prototypes of each category.
>
> 2) The switching mechanism under the one-for-all setting will classify each input image into a single category and choose the corresponding codebook and expert for reconstruction. For the normal images, it is highly likely to be classified into the correct category and thus switch the proper reconstruction expert and codebook. For the abnormal images, there remains big uncertainty that which reconstruction expert and codebook will be switched, because the anomalies are unseen during training. Thus, the reconstruction uncertainty of the abnormal image is increased.
> Noting that the difference between normal and abnormal is the key factor deciding the anomaly detection performance. Thus, this uncertainty of anomalous could improve the performance under the one-for-all setting, which also proves that the switching mechanism of our proposed method is more suitable for multi-class anomaly detection.
>
> The reason for the significant performance differences:
>
> The performance differences in different categories are due to:
> i) the data distribution of each category is different, corresponding to different requests for model representation ability.
> ii) our network architecture is simply implemented without many tricks, such as hyperparameter tuning for each category (\eg. prototype numbers, latent dimension) or adaptive architecture for individual categories. On the whole, it gets efficient performances under both the one-for-all and one-for-one settings.
> However, this insightful observation refers to a common phenomenon for all those one-for-all methods, please refer to UniAD. Even for the one-for-one method, such as the well-known CutPaste, its performance on cable is only 81.2 due to the complex and noisy data distribution.

---

> > ### Author Response · Authors · 2023-08-18
> >
> > Dear Reviewer ksQL,
> >
> > We deeply appreciate your thoughtful review and your time. Following your constructive suggestions, we have discussed the performance  saturation and explained the performance gaps under one-for-one setting with extra experimental results, and revised the descriptions and figures.
> >
> > We have tried our best to address the mentioned concerns/problems in the rebuttal. We would like to know if you have anything unclear or so. Please feel free to let us know. We are delighted to clarify them.
> >
> > If our response has addressed your concerns, would you mind considering re-evaluating our work based on the updated information?
> >
> > Best regards, Authors

---

> > ### Comment · Reviewer_ksQL · 2023-08-20
> > **I keep my initial decision!**
> >
> > After carefully read the rebuttal, I think this paper which with the problems of technical flaws, weak evaluation, inadequate reproducibility and incompletely addressed ethical considerations, are still unsolved.

---

> > ### Comment · Reviewer_ksQL · 2023-08-20
> > **The decision is keeping consistent.**
> >
> > The method is a stacked work with the previous techniques, the experimental results and analysis are not attracting.

---

> > > ### Author Response · Authors · 2023-08-20
> > >
> > > Thanks for your reply.
> > >
> > > We sincerely hope the reviewer ksQL could **concretely point out** the technical flaws, evaluation weakness, inadequate reproducibility, as well as the ethical considerations. First, it is really difficult for us to figure out what is the **ethical problem** in anomaly detection, as a wide-studied filed. Furthermore, we have enough confidence to **firmly uphold our reproducibility**, as our code  has been attached in the supplementary files and will be released to public. In addition, we have demonstrate exhaustive experiments on three public datasets, ie. MVTec-AD, CIFAR-10 and VisA, compared to ten methods published in recently two years (after 2021). The competetive experimental results and intrinsically visualization could verify our technical soundness.
> > >
> > > As for the concern of stacked work with the previous techniques, we hope to resolve it from the following aspects:
> > >
> > > 1) Our model is problem-oriented and elaborately-designed to alleviate the pivotal `identical shortcut' issue in anomaly detection. In contrast to most previous methods that model the continuous latent space and suffer from good generalizability to anomalies, we aim to model discrete space to intrinsically prevent anomaly information leakage into reconstruction. However, the existing memory-based methods recombine and aggregate the discrete memory items, falling into an unknown continuous latent space which might be distorted. In contrast, we force the anomaly features to be replaced by a single discrete prototype. VQ is a proper pathway to optimise prototypes for crimping information bottleneck. Rather than directly employ the original vector quantization mechanism, we elaborately investigate a cascaded VQ transformer to overcome the "prototype collapse" problem, which could also reduce the decoding search time and retain high inference speeds. This designation fits for the hierarchical nature of vision, and matches to calibration of the anomaly score with hierarchical prototype-oriented optimal transport. As far as we know, this is the first try to impose strict restrictions on discrete latent space, and it is the first time to verify the validity of VQ for anomaly detection.
> > >
> > > 2) The optimal transport learning is not only developed to facilitate prototype learning as the previous works, but also dexterously measure the feature level anomaly score to robustly and accurately identify anomalies.
> > >
> > > 3) Our model targets for the challenging 'one-for-all' case, which suffers from the 'identical shortcut' issue more severely, as the model generalizability increases due to the complex data distribution of multiple classes. To tightly fit the unified case, we propose the switching mechanism to choose corresponding codebook and expert for different data distribution, thus alleviating the `identical shortcut' issue.
> > >
> > > In the context of visual anomaly detection, we believe that our method provide a new way  to model discrete space to intrinsically prevent anomaly information leakage into reconstruction, constitutes meaningful contributions.

---

### Official Review · Reviewer_XjaL · 2023-07-09

**Soundness:** 3 good
**Presentation:** 3 good
**Contribution:** 3 good
**Rating:** 5
**Confidence:** 4

**Summary:**

This paper proposes a feature reconstruction based framework for multi-class anomaly detection, called hierarchical vector quantized Transformer (HVQ-Trans). To address the "identical shortcut" problem occurring in the reconstruction-based framework, the proposed method replaces the original encoding features with the nearest iconic prototypes learned from normal training data, and then decoded with a VQ-based transformer decoder to reconstruct the anomaly regions into normal regions. Besides, a hierarchical prototype-oriented learning and anomaly scoring method is developed to guide prototype learning and accurately identify anomalies.




**Strengths:**

1. The motivation is clear and reasonable
2. The writing quality and paper structure are good.
3. The idea of using vector quantization for feature reconstruction in anomaly detection is interesting. The proposed method is technically sound.
4. The proposed method obtains decent performance improvement than the current methods.

**Weaknesses:**

1. Using prototypes may lose high-frequency information leading to imprecise feature reconstruction.

2. Is the optimal transport necessary? How about only simply using similarity scores as the weights?

3. The proposed HVQ-Trans has large model parameters. Please compare the proposed method with UniAD with the inference speed and model parameters.

4. Please add more visualization comparisons with related methods, such as UniAD.

**Questions:**

see the weaknesses

**Limitations:**

see the weaknesses

---

> ### Author Rebuttal · Authors · 2023-08-09
>
> Thanks a lot for your positive comments for this submission! We have tried our best to address the mentioned concerns in the rebuttal. Feel free to let us know if there is anything unclear or so. We are happy to clarify them.
>
> A1: Yes. We want to emphasize that the different degrees of imprecise reconstruction between normal and anomaly are the key factor in anomaly detection. In other words, we pursue the difference in reconstruction ability between normal and anomaly, rather than imprecisely reconstruct them.
>
> In specific, during training, the typical normal patterns are recorded in the discrete variables, i.e., iconic prototypes. When encountering the anomalous during the testing stage, the abnormal patterns will also be quantized as the normal prototypes, leading to larger feature migration and information loss, highlighted by higher reconstruction error. It is worth noting that while information loss triggered by VQ is exist for normal images, it is further significantly pronounced for anomaly images. Thus, this discrepancy in information loss serves as a key factor in effective anomaly detection. By investigating this difference, we can enhance the accuracy of our model in distinguishing abnormal regions.
>
> A2: We employ optimal transport (OT) for better exploring relationships between learnable prototypes and visual features of input images. Specifically, traditionally used similarity scores such as Euclidean or Cosine only concern the point-wise relationship between two sets and pose equally important prior to any such kind of relationship. In practice, however, it is usually sub-optimal to introduce such non-informative prior and leads to poor performance. In our model, optimal transport learning is not only developed to facilitate prototype learning with the regularization on the feature distribution level, but also dexterously measure the feature level anomaly score to robustly and accurately identify anomalies.
>
> Moreover, as we reported in Table 4 of the main paper, we can see the POT module leads to 0.4\% and 0.1\% performance gain on anomaly detection and localization, respectively, on the MVTec-AD dataset.
> Furthermore, as shown in Table 1 in the appendix, the POT module results in 2.6\% improvements on the CIFAR-10 dataset. This demonstrates the POT module is effective to detect anomalies due to its well-established measurement alignment, especially showing the stability for the complex scenarios of CIFAR-10.
>
> A3: The comparison results are listed in Table 2 of the uploaded PDF. With the image size fixed as $224 \times 224$, we compare our model with all competitors regarding the inference FLOPs and learnable parameters. We can tell that the advantage of our approach does not come from a larger model capacity. This table will be included in the camera ready.
>
> A4: The visualization comparisons with UniAD are shown in the unloaded PDF. More visualization comparisons will be added to the appendix of our revised paper.

---

### Official Review · Reviewer_Ah19 · 2023-07-13

**Soundness:** 3 good
**Presentation:** 3 good
**Contribution:** 3 good
**Rating:** 5
**Confidence:** 5

**Summary:**

This paper introduces a novel approach to multi-class anomaly detection (AD) by integrating hierarchical embedding vector quantization. To tackle the problem of identical shortcuts in the reconstruction-based AD paradigm, the authors suggest to enlarge abnormality's reconstruction residue by introducing discrete prototypes in the model. Additionally, the model incorporates a switching mechanism to further improve the feature reconstruction process. The proposed model is evaluated on MVTec and CIFAR-10 datasets and compared against prior arts.

**Strengths:**

1. The paper is well-written and easy to follow.

2. The hierarchical vector quantized transformer is well presented. Appendix about the loss back-propagation explains parameter updates regardless quantization operation.

3. Extensive experiments are conducted. Ablation studies on model components suggest effectiveness of the proposed method.

**Weaknesses:**

1. The vector quantization mechanism described in this paper appears to be a specific instance of the memory mechanism, where a continuous-valued feature is substituted with a numerical prototype. If this is indeed the case, such a memory mechanism has been widely employed in previous studies on AD.

2. What does 'C' represent in Figure 2 and Figure 3? Is it indicating feature concatenation? The paper briefly mentions feature aggregation but does not provide a specific definition or explanation of the feature aggregation operation.

3. Within the switching mechanism, a multi-category classifier, N codebooks, and reconstruction experts are necessary. In this scenario, can we consider the model as a combination of a single encoder and N decoders that are specific to each class?

4. The experimental evaluation compares the proposed method with US, PSVDD, PaDim, CuPaste, MKD, and DREAM. However, there exist several recently proposed AD models, such as Patchcore [1], RD4AD [2], CS-Flow [3], UTRAD [4], which outperform the aforementioned methods. It is highly recommended to include these methods in the experimental analysis as well.

[1] Karsten Roth et al. “Towards total recall in industrial anomaly detection”. Proceedings of the IEEE/CVF Conference on Computer Vision and Pattern Recognition. 2022, pp. 14318– 14328.
[2] Hanqiu Deng and Xingyu Li. “Anomaly Detection via Reverse Distillation from One-Class Embedding”. Proceedings of the IEEE/CVF Conference on Computer Vision and Pattern Recognition. 2022, pp. 9737–9746.
[3] Marco Rudolph et al. “Fully convolutional cross-scale-flows for image-based defect detection”. Proceedings of the IEEE/CVF Winter Conference on Applications of Computer Vision. 2022, pp. 1088–1097.
[4] Liyang Chen et al. “UTRAD: Anomaly detection and localization with U-Transformer”. Neural Networks 147 (2022), pp. 53–62.

**Questions:**

Please refer to the weakness section. Thanks.

**Limitations:**

No discussion on limitations and potential negative social impact.

---

> ### Author Rebuttal · Authors · 2023-08-09
>
> Thanks a lot for your positive comments! We have tried our best to address the mentioned concerns/problems. Feel free to let us know if there is anything unclear or so. We are happy to clarify them.
>
> A1: To some extent, our vector-quantized prototype can be regarded as a special kind of memory item.
> As the reviewer mentioned, a branch of approaches~\cite{roth2022towards,gong2019memorizing,xiang2021painting} have investigated the memory-augmented networks recently, which augment the deep autoencoder with a memory module to record the normal information of training data. These methods hope to obtain low reconstruction error for normal samples and highlight the reconstruction error if the input is not similar to normal data, that is, an anomaly. The relevant memory items are retrieved and weighted averaging all the related memory content are aggregated into the decoder for reconstruction.
>
> Thus, we claim three differences between our proposed method with the previous methods:
> 1) The discrete memory items are recombined and weighted averaged in previous works, falling into an unknown continuous latent space that might be distorted. Intuitively, some anomalous regions can not be reconstructed by the discrete latent memory but could be decoded from the unknown latent space. In contrast, we set a strict information bottleneck to enforce the abnormal data point flipping to a normal data point, constraining the leakage of abnormal information.
> 2) The prototypes are individually learned for each category, and adaptively chosen by a switching mechanism. This mechanism tightly fits the unified case, which suffers from the `identical shortcut' issue more severely, as the model generalizability increases due to the complex data distribution of multiple classes. We choose specific codebooks for different data distributions to alleviate the 'identical shortcut' issue.
> 3) A hierarchical VQ-based approach is developed to better overcome the "codebook collapse" problem and effectively merge fine-grained concrete information with abstract semantics to maximize the nominal information available.
> 4) In addition, we introduce the POT module to guide the learning process of prototypes with the help of OT theory, to facilitate prototype learning and dexterously measure the feature level anomaly score. On the contrary, the existing memory-based methods always use cosine similarity or Euclidean distance to learn memory vectors.
>
> A2: Yes, the `C' represents the feature concatenation.
>
> The feature aggregation includes two steps: Firstly, the visual tokens are concatenated with the global quantized vector $\theta$; Secondly, a layer-wise embedding function $\Upsilon^l(\cdot)$ is utilized for feature fusion. Then the Quantization is performed on the aggregated features.
> Specifically, we fuse the multi-level visual tokens $h^{l-1}$ with the global quantized vector $\theta$ to learn hierarchical prototypes, maximizing the preserved nominal information, stated as $\texttt{Quantize}(\Upsilon^l\left(\left[ h^{l-1}, \theta \right]\right)) = e_k^l$. Here, $\left[ \cdot \right]$ denotes the concatenation operation, and $\Upsilon^l(\cdot)$ refers to the embedding function. We replace the fused feature with its most similar prototype $e_k^l$ in the codebook as the corresponding quantized vector.
>
> A3: The switching mechanism did contain a multi-category classifier, N codebooks, and N reconstruction experts. However, the reconstruction experts are only the final part of our HVQ-Transformer decoder.
>
> As shown in Figure 2 in the main paper, the HVQ-Transformer has four graded decode layers. In each decoder layer, the refined queries $q^l$ from the previous layer are crossly connected to the quantized normal prototypes $z^l$, through the multi-head cross-attention layer. Hence, the values at abnormal regions of $q^l$ will be suppressed, and the abnormal signals could be rarely transmitted for reconstruction. The switching reconstruction experts are placed after the four decoder layers, for flexible feature reconstruction. Specifically, the visual tokens from the last \textit{VQTrans-dec} layer are depicted as $d^0 \in \mathbb{R}^{N \times C}$, which is expected to reconstruct the patch features as $\Psi_m(d^0)$,
> where $m^{th}$ expert network $\Psi_m$ is selected for reconstruction.
>
> A4: We replenish the experiments compared to all the mentioned methods, as shown in Table 1 of the uploaded PDF. The results show that we achieve superior performances.
>
> Limitations: The limitation and potential negative social impact are presented in the 'discussion' part of the conclusion section.

---

> > ### Comment · Reviewer_Ah19 · 2023-08-19
> > **Thanks for the feedback from authors.**
> >
> > The authors answer most of my questions. But I am still concerning the innovation of the vector quantization mechanism. So I keep my original rating.

---

> > > ### Author Response · Authors · 2023-08-20
> > >
> > > Dear reviewer,
> > >
> > > Thank you a lot for your effort in reviewing this submission!
> > >
> > > We cautiously wish to assert the innovation of our hierarchical vector quantized (HVQ) Transformer. Rather than directly employ the original vector quantization mechanism, we elaborately investigate a cascaded VQ transformer to overcome the "prototype collapse" problem, which could also reduce the decoding search time and retain high inference speeds. This designation fits for the hierarchical nature of vision, and matches to calibration of the anomaly score with hierarchical prototype-oriented optimal transport. Even though HVQ-Trans is based on vector quantization mechanism, it achieves significantly better anomaly detection performance than the naive VQ model and the previous memory-base algorithms. Please consider that the paper also provides novel techniques, such as switching mechanism and the prototype-oriented learning and scoring, which deliver stable training and improved performance.
> > >
> > > In addition, we would like to highlight that simplicity is the ultimate form of sophistication. VQ is a proper pathway to optimize prototypes for crimping information bottleneck, which is an effective way to achieve our purpose rather than the novelty itself. In the context of visual anomaly detection, we firmly believe that our method provide a new way to model discrete space without confusion (recombine or aggregation), which intrinsically prevents anomaly information leakage into reconstruction, constitutes meaningful contributions.
> > >
> > > Thanks again for your valuable time! We are genuinely grateful for your response and help.
> > >
> > > Best regards,
> > >
> > > Anonymous author

---

### Official Review · Reviewer_JdWV · 2023-07-19

**Soundness:** 3 good
**Presentation:** 3 good
**Contribution:** 2 fair
**Rating:** 5
**Confidence:** 4

**Summary:**

The paper proposes using hierarchical vector-quantized transformer-based autoencoders for image anomaly detection. The key contribution is the use of prototypes learned using an optimal transport algorithm.

**Strengths:**

The paper addresses a critical problem in reconstruction-based anomaly detection (good reconstruction of both normal samples and anomalies) and proposes a reasonably solid solution based on discrete latent prototypes.

**Weaknesses:**

1) The first and foremost problem with the proposed approach is the claim that it is "unsupervised". As the paper admits in the final discussion paragraph on page 9, the current work assumes that the category labels are available during the training stage. In that case, how can it be claimed that the proposed method is unsupervised? Also, this may make comparison with unsupervised approaches such as UniAD unfair.

2) The second critical issue is the novelty of the proposed solution because it combines some well-known ideas such as vector-quantized VAE [21] and hierarchical VQ-VAE [17] and some recent ideas such as learning latent prototypes for autoencoders using optimal transport [A, B].

[A] Bie et al., "Learning Discrete Representation with Optimal Transport Quantized Autoencoders", 2023
[B] Oliveira et al., "Improving Variational Autoencoders Reconstruction Using Prototypes", 2023.

However, in my opinion, the paper smartly combines these known ideas and makes some marginal improvements to implement a transformer-based VAE model.

3) The switching mechanism and mixture-of-experts concepts on page 4 have not been clearly explained. In particular, it is not clear what the so-called "classifier for producing logits" and "expert" mean and how these components tie in with the POT module described subsequently. Furthermore, the ablation study talks about "codebook switching" and "expert switching", but these are never described in the paper.

4) Though the paper claims that the proposed method achieves SOTA results, it is not obvious if this is true. Except for UniAD (published in NeurIPS 2002), all the other baseline methods selected for benchmarking are from 2020 or 2021. A cursory glance at more recent works such as [C, D] indicates that the reported performance results fall short of SOTA results.

[C] Liu et al., "SimpleNet: A Simple Network for Image Anomaly Detection and Localization" CVPR 2023
[D] Tien et al., "Revisiting Reverse Distillation for Anomaly Detection", CVPR 2023

5) The calibration of anomaly score based on POT has been emphasized many times, but there appears to be no experiment to demonstrates the importance of this idea.

6) It is not clear how to determine the number of prototypes required? Will this depend on the number of classes (or number of anomaly types)? Why does the performance start decreasing when the number of prototypes start increasing beyond 512?

7) There is no mention about the computational complexity of the proposed approach.


**Questions:**

Please see weaknesses.

**Limitations:**

All the limitations of the proposed method have not been discussed and addressed. The paper does not appear to have any potential negative social impact.

---

> ### Author Rebuttal · Authors · 2023-08-09
>
> A1: Thanks!
> 1) The unsupervised AD methods only utilize the normal images during training stage, while both the abnormal and normal images are utilized at the testing stage. Thus, the 'unsupervised' in AD commonly refers to inaccessible to any abnormal data as supervision at the training stage.
> 2) The category labels are accessible in advance and UniAD also claims that ``The category labels may help the model better fit multi-class data. How to incorporate the unified model with category labels should be further studied''. Thus, in this work, we make full use of category information to better serving our model. Furthermore, we replenish an experiment without category information (no switching mechanism), resulting in 97.4 and 97.2 for  anomaly detection and location, which surpass UniAD.
> 3) Under the one-for-one case, the category information is accessible to all the comparison models, including UniAD.
> Under this case, our model surpass UniAD (the SOTA method for unified case) by 0.3\% and 0.5\% for anomaly detection and localization.
>
> A2: Our model is problem-oriented and elaborately-designed to alleviate the pivotal `identical shortcut' issue in anomaly detection.
> 1) In contrast to most previous methods that model the continuous latent space and suffer from good generalizability to anomalies, we aim to model discrete space to intrinsically prevent anomaly information leakage into reconstruction. However, the existing memory-based methods recombine and aggregate the discrete memory items, falling into an unknown continuous latent space which might be distorted. In contrast, we force the anomaly features to be replaced by a single discrete prototype. Therefore, VQ is a proper pathway to optimise prototypes for crimping information bottleneck. Although VQ-VAE is a well-known technique, it rightly fits our motivation. As far as we know, this is the first try to impose strict restrictions on discrete latent space, and it is the first time to verify the validity of VQ for anomaly detection.
> 2) Moreover, the optimal transport learning is not only developed to facilitate prototype learning as the previous works, but also dexterously measure the feature level anomaly score to robustly and accurately identify anomalies.
> 3) The hierarchical structure is originally designed by merging fine-grained concrete information with abstraction-level semantics, which could avoid the issue of codebook collapse, as shown in Table 5.
> 4) In addition, our model targets for the challenging 'one-for-all' case, which suffers from the 'identical shortcut' issue more severely, as the model generalizability increases due to the complex data distribution of multiple classes. To tightly fit the unified case, we propose the switching mechanism to choose corresponding codebook and expert for different data distribution, thus alleviating the `identical shortcut' issue.
>
> A3:
> 1) The switching mechanism contains a multi-category classifier, $M$ codebooks, and $M$ reconstruction experts. The multi-category classifier takes the image feature as input and output the classification probability (logit) over $M$ category. In order to fit the data diversity property in one-for-all setting, we switch individual codebook (including a group of prototypes) from $M$ codebooks, according to the classification probability. Furthermore, we switch individual reconstruction network (expert) for decoding features from the last decoder layer to re-build the input features from the pre-trained EfficientNet.
> 2) As for POT, it is used to regularize the relationships between features and vector-quantized prototypes, making them better matched compared with Euclidean or Cosine distance used in original VQ-VAE. Therefore, POT only takes effects with features at hierarchical layers.
> 3) 'codebook switching' and 'expert switching' all refer to the action that classifier choose corresponding codebook or reconstruction network.
>
> A4: We replenish the experiments compared to the mentioned methods in Table 1 of the uploaded PDF. The results show that we achieve superior performances.
>
> A5: As we reported it in Table 4 of the ablation study, we can see the POT module leads to 0.4\% and 0.1\% performance gain on anomaly detection and localization, respectively, on MVTec-AD dataset. Furthermore, as shown in Table 1 in appendix, the POT module results in 2.6\% improvements on CIFAR-10 dataset. This demonstrate the POT module is effective to detect anomalies due to its well-established measurement alignment, especially showing the stability on the complex scenarios of CIFAR-10.
>
> A6: The prototypes are grouped into $M$ categories. We only decide the number of prototypes $K$ per group, which is related to the complexity of the data distribution. For the category of texture or simple object, such as bottle and grid, the iconic normal patterns are limited. The corresponding number of prototypes can be set smaller than those complex objects.
>
> Without exhausted parameter tuning for individual category, we set the same number of prototypes for each category. We found that 512 is an empirically proper setting, which performs averagely decent on all the 15 categories. This might because: i) The smaller number of prototypes can't cover all the iconic normal patterns for the complex objects, leads to poor reconstruction ability; ii) Overmuch prototypes will cause the abundance and repetition of iconic normal patterns, which is harmful to model training and disturbs the precise vector quantization. Specifically, redundant prototypes may never be used and optimized during training. At inference, those non-optimized prototypes may be closer with anomaly patterns, leading to leakage of abnormal information, and finally resulting in poor anomaly detection. Moreover, it's worth noting that different numbers of prototypes can consistently surpass all the competitors in Table 1 and Table 2.
>
> A7: We replenish the computational complexity in Table  2 of the uploaded PDF.

---

> > ### Comment · Reviewer_JdWV · 2023-08-13
> > **Response to Author Rebuttal**
> >
> > I thank the authors for their detailed response, which address some of my core concerns (regarding the switching mechanism and unsupervised claim). Hence, I'm inclined to increase my rating.

---

> > > ### Author Response · Authors · 2023-08-18
> > >
> > > Thank you for your valuable suggestions that help us improve the manuscript. We are glad that you increase your rating for this paper.
> > > Thanks again for your effort in reviewing this submission!

---

### Author Rebuttal · Authors · 2023-08-10

Dear reviewers and AC

Thanks a lot for your effort in reviewing this submission! We have tried our best to address the mentioned concerns/problems in the rebuttal. Feel free to let us know if there is anything unclear or so. We are happy to clarify them.

Best, Authors

---

### Decision · Program_Chairs · 2023-09-21

**Decision:**

Accept (poster)

**Comment:**

The paper introduces an approach based on hierarchical vector-quantized transformer autoencoders for image anomaly detection. The manuscript received ratings of: borderline accept, borderline accept, reject, borderline accept and borderline accept. While the reviewers appreciated the motivation, idea and paper writing, they also raised some concerns such as, lack of clarity regarding the switching mechanism, comparison with more recent AD models, parameter comparison and performance under one-for-one setting and one-for-all setting. Authors responded to reviewer's queries by submitting a rebuttal. The rebuttal largely addressed the initial concerns raised by the reviewers. While four reviewers were on the positive side, one reviewer remained on the negative side. Given that four reviewers are generally positive about the paper as well as author's rebuttal (specifically justifying the performance comparison under one-for-all setting), the recommendation is accept. Authors are strongly encouraged to take into account the suggestions of the reviewers as well as changes acknowledged in the rebuttal when preparing the final manuscript.